# COLORISTANET FOR PHOTOREALISTIC VIDEO STYLE TRANSFER

## ABSTRACT

Photorealistic style transfer aims to transfer the artistic style of an image onto an input image or video while keeping photorealism. In this paper, we think it's the summary statistics matching scheme in existing algorithms that leads to unrealistic stylization. To avoid employing the popular Gram loss, we propose a self-supervised style transfer framework, which contains a style removal part and a style restoration part. The style removal network removes the original image styles, and the style restoration network recovers image styles in a supervised manner. Meanwhile, to address the problems in current feature transformation methods, we propose docouple instance normalization to decompose feature transformation into style whitening and restylization. It works quite well in ColoristaNet and can transfer image styles efficiently while keeping photorealism. To ensure temporal coherency, we also incorporate optical flow methods and ConvLSTM to embed contextual information. Experiments demonstrates that ColoristaNet can achieve better stylization effects when compared with state-of-the-art algorithms.

## 1 INTRODUCTION

Nowadays rapid development of video-capture devices has made videos become a mainstream information carrier (Hansen, 2004). People usually post videos accompanied with different color styles on social media (Kopf et al., 2012; Xu et al., 2014) to share daily life, express different emotions, and get more exposures (Yan et al., 2016; Zabaleta & Bertalmío, 2021). Thus, photorealistic video style transfer or automatic color stylization becomes popular in many mobile devices. Different from artistic style transfer (Gatys et al., 2016; Huang & Belongie, 2017), photorealistic video style transfer or automatic color stylization needs to replace color styles in original videos with one or multiple reference images and keep the outputs maintain "*photorealism*". The photorealism in style transfer refers to that stylization results should look like real photos taken from cameras without any spatial distortions or unrealistic artifacts. Moreover, algorithms need to run in realtime.

Several popular algorithms have been proposed to conduct photorealistic style transfer for single image. DeepPhoto (Luan et al., 2017) incorporated semantic segmentation masks to guide style transfer and utilized a photorealism regularization term to reduce spatial distortions. PhotoWCT (Li et al., 2018) exploited whitening and coloring transforms (WCT (Li et al., 2017c)) to conduct arbitrary style transfer and used photorealistic smoothing to remove spatially inconsistent stylization. $WCT^2$ (Yoo et al., 2019) proposed a wavelet corrected transfer based on WCT to preserve structural information while stylizing images at the same time. PhotoNAS (An et al., 2020) proposed a neural architecture search framework for photorealistic style transfer and achieved impressive results.

Although these algorithms can conduct style transfers in many scenarios, their stylization results still contain unpleasant artifacts or look unreal, and some algorithms need additional supports. In Figure 1 (a), given a content image which contains a tree in autumn and a style reference, previous state-of-the-art algorithm $WCT^2$ (Yoo et al., 2019) will generate synthesized images with obvious structural artifacts. Besides, these algorithms conduct style transfer by matching the summary statistics of content features with style references completely, which will lead to unrealistic stylization as in Figure 1 (b). For photorealistic style transfer in videos, there are only very few existing algorithms that can only perform style transfer with constraints. MVStylizer (Li et al., 2020) need good stylization initilaization at the first frame and Xia's method (Xia et al., 2021) incorporates additional semantic masks for each frame in videos. These problems limit these methods' usage in many real applications.

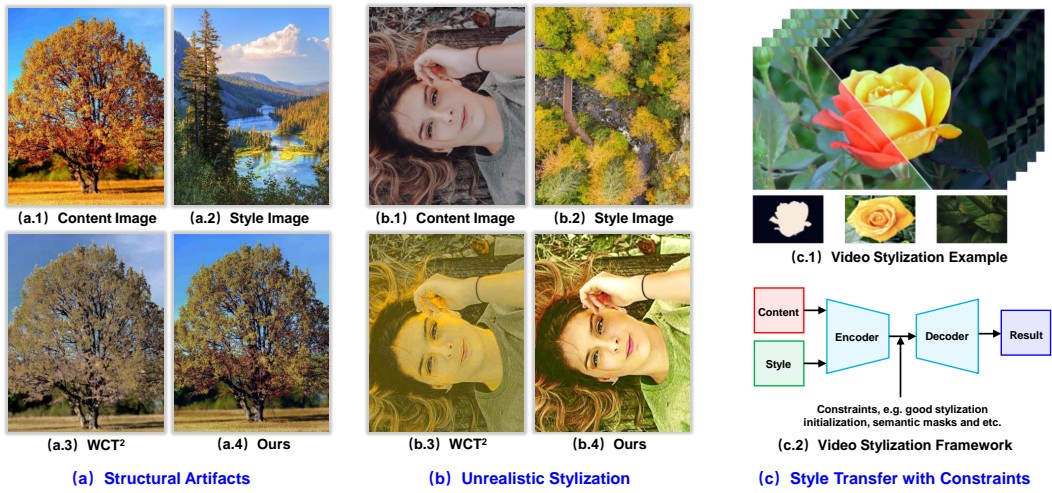

Figure 1: Illustration of unsolved problems in photorealistic style transfer. From left to right: (a) Previous state-of-the-art algorithm WCT$^2$ (Yoo et al., 2019) generates stylization results with obvious structural artifacts. (b) The stylization result produced by WCT$^2$ (Yoo et al., 2019) looks painterly and slightly unreal. (c) Video stylization algorithms need additional inputs, such as good stylization initialization (Li et al., 2020) or semantic masks (Xia et al., 2021), to guide style transfer.

In this paper, we aim to solve the problems listed above in photorealistic video style transfer. Different from previous algorithms which match summary statistics of content images to that of style references through whitening and coloring transformation (Li et al., 2018), adaptive instance normalization (An et al., 2020) and the Gram loss (Luan et al., 2017), we propose a style removal and restoration framework in a self-supervised manner to conduct arbitrary style transfer while keeping photorealism. Our motivation is that during photorealistic style transfer, if we can remove the style of image content without destroying image structures, we can recover its original style by using the content image both as style reference and stylization target. According to our experiences, artifacts produced by PhotoWCT (Li et al., 2018), WCT$^2$ (Yoo et al., 2019), and PhotoNAS (An et al., 2020) come from two parts: (1) the Gram loss; (2) whitening and coloring transformation (WCT (Li et al., 2017c)). In our method, we avoid using the Gram loss and train networks with the content loss only (Gatys et al., 2016). We improve the summary statistics matching scheme with decoupled instance normalization which can remove original image styles and add new styles for inputs without hurting image structures. Meanwhile, decoupled instance normalization does not match styles of reference images completely and avoid unrealistic stylization in Figure 1 (b). To keep temporal consistency in videos, we exploit optical flow estimation (Teed & Deng, 2020) and ConvLSTM (Shi et al., 2015a) to conduct consecutively style transfer. We summarize our contributions as follows:

• In this paper, we propose a novel photorealistic video style transfer network called ColoristaNet, which can conduct color style transfer in videos without introducing painterly spatial distortions and inconsistent flickering artifacts. We put many videos in the supplementary material to compare with other state-of-the-art algorithms.

• We propose decoupled instance normalization which works together with ConvLSTM (Shi et al., 2015a) to implement structure-preserving and temporally consistent feature transformation. The decoupled instance normalization decomposes style transfer into feature whitening and stylization, which can avoid unrealistic style transfer.

• ColoristaNet can adapt color styles in videos consecutively with multiple different style references and runs faster than most of recent algorithms. Qualitative results and a user study show that our method outperforms other state-of-art algorithms in making a balance between good stylization results and photorealism. Besides, we also conduct extensive ablation studies whose results demonstrate the effectiveness of different modules and designs in ColoristaNet clearly.

## 2 PRELIMINARIES AND MOTIVATIONS

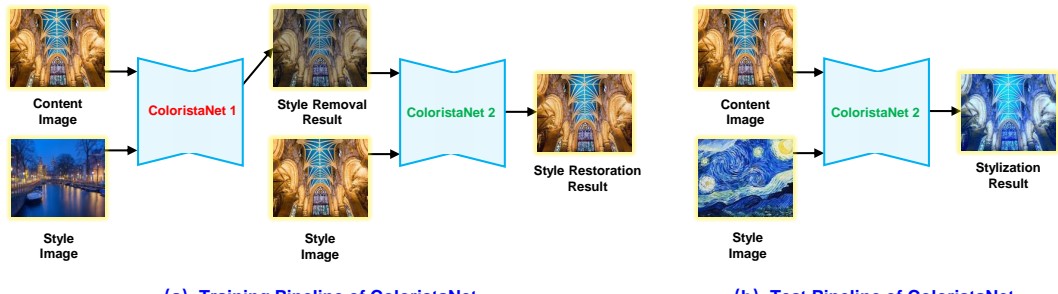

Figure 2: The training and test pipeline of ColoristaNet. During training, a ColoristaNet is firstly exploited to replace the color style of a content image with a style reference. Then, another ColoristaNet restores the style of the synthesized image by using the original content input as a style reference. During testing, the style restoration ColoristaNet can conduct style transfer efficiently.

Neural style transfer algorithms (Gatys et al., 2016; Li et al., 2017b) have achieved great success in creating artistic images of high perceptual quality. Using neural representations to separate and recombine content and style of arbitrary images is widely investigated and adopted by researchers (Li et al., 2017a; Zhu et al., 2017b; Johnson et al., 2016; Ledig et al., 2017). In Gatys' paper, the Gram matrix consists of the correlations between different filter responses and describe the overall image style, and features in deeper layers is thought capturing the high-level content in term of objects and their arrangement. Then style transfer problems can be solved by matching summary statistics of content inputs to that of style references. However, although such a framework works quite well for artistic style transfer, it is not suitable for photorealistic style transfer. Because matching the summary statistics of content images with arbitrary style references will generate unpleasant artifacts or distortions. Photorealistic style transfer algorithms, such as DeepPhoto (Luan et al., 2017), PhotoWCT (Li et al., 2018), $WCT^2$ (Yoo et al., 2019), and PhotoNAS (An et al., 2020), focus on eliminating artifacts or distortions with additional smoothing term or other regularization terms. As shown in Figure 1, structural artifacts and unrealistic stylization is hard to be avoided.

In this paper, we hold an assumption that in previous methods, it's the summary statistics matching scheme in learning objectives and feature transformation modules that lead to structural artifacts or unrealistic stylization. That means the Gram loss (Gatys et al., 2016), AdaIN (Huang & Belongie, 2017)] and WCT (Li et al., 2018) are problematic in photorealistic style transfer. To address these issues, we propose ColoristaNet with: (1) a self-supervised style transfer framework that avoids employing the Gram loss during training; and (2) a novel feature transformation module to substitute AdaIN or WCT to perform summary statistics matching. For the self-supervised style transfer framework, as shown in Figure 2, if we can remove the style of an image without hurting its structure, the style restoration problem becomes a fully supervised one. The reason why our idea works is because in the photorealistic setting, image structures are shared and unchanged during style transfer. The benefits of our self-supervised learning scheme come from two folds: (1) We avoid employing the Gram loss or other regularization loss functions that will result structural artifacts or blur effects; (2) Our learning targets are real photos which can ensure that the stylization results make a good balance between stylization and photorealism. We discuss more details in Appendix B.1.

To address the problems brought by AdaIN and WCT, ColoristaNet incorporates a novel feature transformation module called decoupled instance normalization (DecoupleIN) to match the styles of content images with that of reference images. DecoupledIN is inspired by AdaIN, and decompose the feature transformation into a style whitening step and a restylization step. This decomposition avoids forcing the feature statistics of content images to match that of style images directly, and show impressive results. In addition, as we conduct video style transfer, we need to keep temporal coherency in consecutive frames. So we employ optical flow methods to estimate pixel locations in a next frame and propagate style information through ConvLSTM (Shi et al., 2015a) as shown in Figure 3. We give more explanations about the design of ColoristaNet in Appendix B.2.

# 3 METHOD

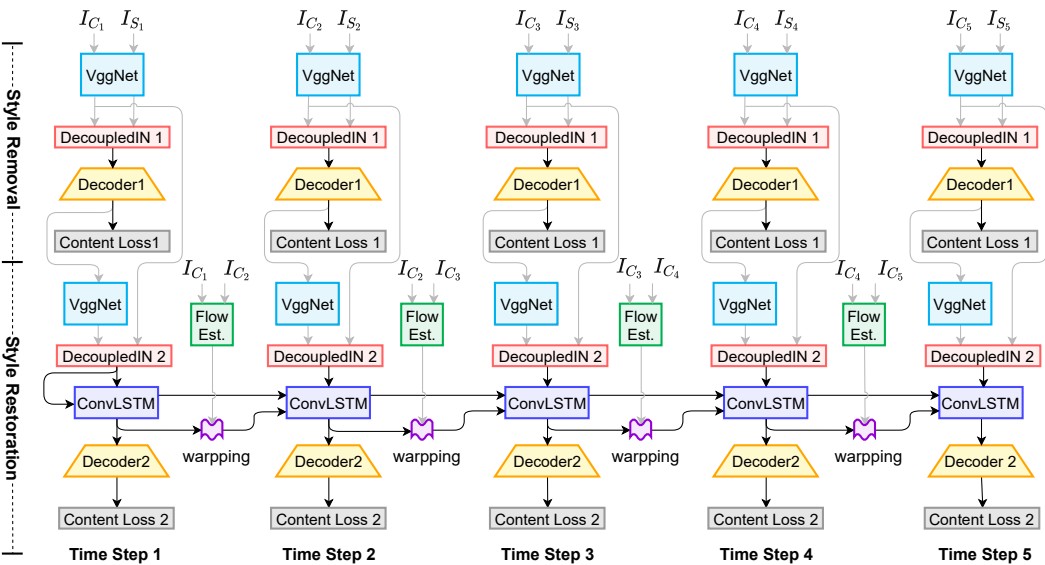

Figure 3: Illustration of the training pipeline of ColoristaNet. There are five content frames and five style references for a video clip. For each frame, it firstly passes through a style transfer network to conduct style removal and then go through another style transfer network for style restoration. In style restoration, features from different time steps are connected with a ConvLSTM unit. A flow estimation network (RAFT (Teed & Deng, 2020) with fixed parameters) predicts optical flow between two adjacent frames to warp the hidden states of ConvLSTM for movement compensation. Note that parameters of style removal and restoration networks at different time steps are shared.

## 3.1 OVERVIEW OF THE PROPOSED METHOD

In this section, we introduce the training pipeline of ColoristaNet. For the test pipeline, we give more details in Appendix A.2. As discussed in the previous sections, we exploits a style removal ColoristaNet and a style restoration ColoristaNet to conduct end-to-end training (as shown in Figure 3). Given a video clip with five image frames, we send them together with randomly selected style references to a style transfer network to conduct style removal. Then these style removal results are used as content inputs and the original inputs are used as style images to perform style restoration with another style transfer network. Both style transfer networks are in a similar structure except that the style restoration incorporates RAFT (Teed & Deng, 2020) and ConvLSTM (Shi et al., 2015a) to keep temporal coherency. For the learning objective, they are trained with two content loss respectively, and their learning targets are original video frames. During the test, the second style transfer network is exploited to conduct style transfer without additional constraints.

## 3.2 STYLE TRANSFER NETWORK

Figure 3 shows the architecture of two style transfer networks roughly. A style transfer network consists of a VGG-19 encoder Simonyan & Zisserman (2014), four decoupled instance normalization modules, four ConvLSTM units Shi et al. (2015b), and a decoder to generate final output. Note that, in style removal, ConvLSTM units are removed, since it doesn't need context information. Given an input image pair $(I_{C_t}, I_{S_t})$, feature maps at "$conv1\_1$", "$conv2\_1$", "$conv3\_1$" and "$conv4\_1$" of a VGG-19 network $\Phi_{vgg}$ with frozen parameters are extracted: These multiscale features pass through four decoupled instance normalization modules (shown in Figure 4) to map feature statistics of the content image to match that of its style reference. In style restoration, RAFT is exploited to estimate pixel locations in adjacent frames and ConvLSTM is responsible to incorporate contextual information.

Then, a U-net Ronneberger et al. (2015) style decoder fuses information across different scales and generate stylization results. We give a very detailed structure configurations in Appendix A.2.

### 3.3 DECOUPLED INSTANCE NORMALIZATION

Matching feature statistics through feature transformation has been proven powerful in both artistic style transfer and photorealistic style transfer (AdaIN (Huang & Belongie, 2017) and WCT (Li et al., 2017c)). But directly applying AdaIN will hurt some subtle image details, and WCT often leads to unrealistic stylization. Here we propose decoupled instance normalization (DecoupledIN) that decomposes the feature transformation into feature whitening and stylization. Figure 4 shows the DecoupledIN module. Given a content input $f_{C_t,i}$ and a style input $f_{S_t,i}$ at $i$-th layer, we remove the style of $f_{C_t,i}$ as Equation 1 firstly, and then send the whitened result $\widetilde{f}_{C_t,i}$ and style input $f_{S_t,i}$ into a $3 \times 3$ convolutional layer with $2c$ filters to conduct AdaIN ($c$ is the number of input feature channel). Finally, we reduce the feature channels of stylized features $g_{t,i}$ to be the same with inputs. The overall process of DecoupledIN can be described with the following equations:

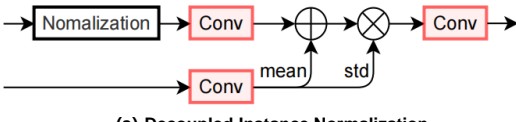

**(a) Decoupled Instance Normalization**

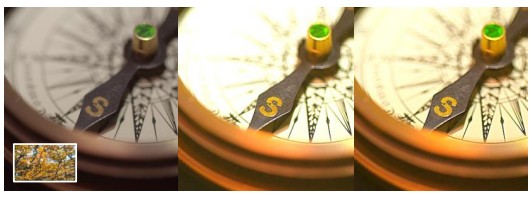

Content and Style     AdaIN     DecoupleIN

**(b) Visual Comparison of AdaIN and DecoupledIN**

Figure 4: Conceptual illustration of decoupled instance normalization and visual comparison with AdaIN (Huang & Belongie, 2017).

$$
\begin{aligned}
\text{Whitening}: f'_{C_t,i} &= \frac{f_{C_t,i} - \mu\left(f_{C_t,i}\right)}{\sigma\left(f_{C_t,i}\right)}, \\
\text{Transform}: f''_{C_t,i}, f''_{S_t,i} &= \text{Conv}\left(f'_{C_t,i}\right), \text{Conv}\left(f_{S_t,i}\right), \\
\text{Stylization}: g_{t,i} &= \sigma\left(f''_{S_t,i}\right)\left(\frac{f''_{C_t,i} - \mu\left(f''_{C_t,i}\right)}{\sigma\left(f''_{C_t,i}\right)}\right) + \mu\left(f''_{S_t,i}\right),
\end{aligned}
\tag{1}
$$

where $\mu$ and $\sigma$ calculate the mean and standard deviation for each feature channel respectively. Figure 4 indicates that style transfer through DecoupledIN will generate better stylization results without hurting image details when compared with using AdaIN. We attribute this to that directly changing the content feature statistics will make some neurons work out of their working range and remove detailed structures. When we separate feature whitening from stylization, we conduct feature transformation more smoothly and get better results. We disucss DecoupledIN and conduct more ablations in Appendix B.2 to investigate our assumptions. Experiments indicates that more whitening will lead to better stylization effects.

### 3.4 LOSS

Unlike previous works which employed multiple loss functions including content loss, temporal loss, Gram loss, and etc, we simply employ content loss to constrain the structure of the stylization results to be the same as content inputs (for style removal) and guide ColoristaNet to generate photorealistic videos like real world videos (for style restoration). Given a stylization result $I_G$ and an input image $I_C$, we use the feature maps at conv4_1 layer of VGG-19 to calculate the content loss. The two ColoristaNets are trained end-to-end without sharing parameters. For a video clip with $N$ frames, the learning objective becomes:

$$
\mathcal{L} = \sum_{i=1}^{N} \mathcal{L}_{content}\left(I_{G_{1,i}}, I_{C_i}\right) + \lambda \mathcal{L}_{content}\left(I_{G_{2,i}}, I_{C_i}\right),
\tag{2}
$$

where $I_{G_{1,i}}$ and $I_{G_{2,i}}$ are generated images of style removal and restoration networks respectively. In experiments, we set $\lambda = 1$ and get impressive results.

# 4 EXPERIMENTS

## 4.1 EXPERIMENTAL DETAILS

We conduct extensive experiments to indicates the effectiveness of the proposed method. Due to the page length limit, we put implementation details into the appendix, including datasets, evaluation protocols and training and test settings (in Appendix A). We also put more results and discussions in the appendix part to compare with state-of-the-art methods and investigate the effectiveness of different designs and modules in ColoristaNet.

## 4.2 QUALITATIVE COMPARISON

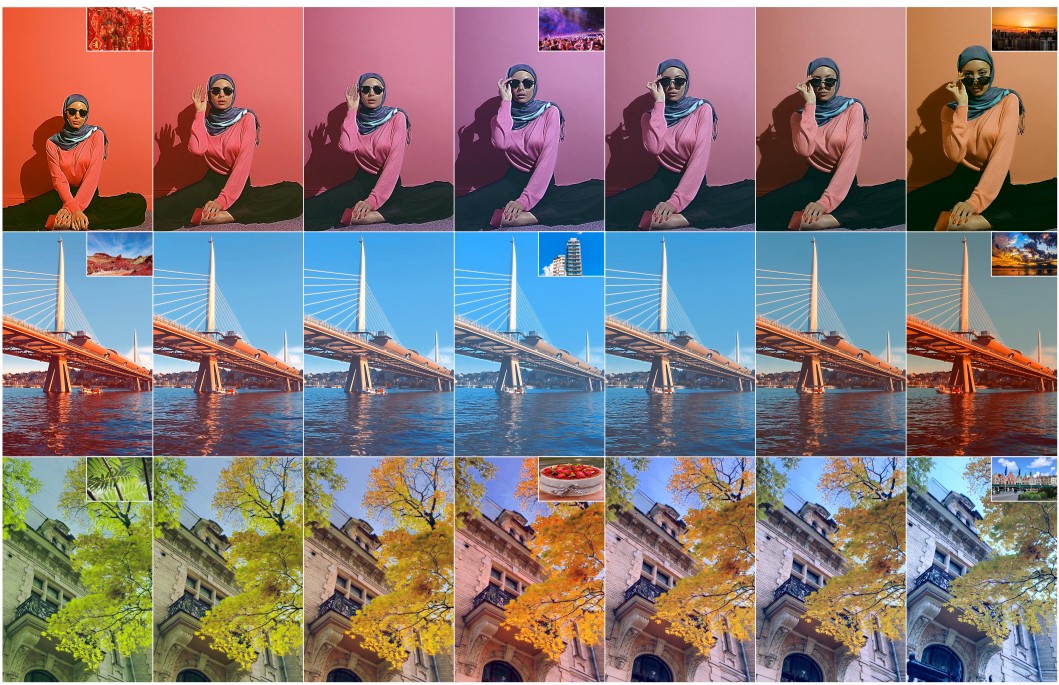

Figure 5: Multiple color stylization on a set of consecutive frames with ColoristaNet. Stylization targets are visualized in the right up corner of first, fourth, and seventh columns. From left to right, color styles of input frames changed smoothly without any painterly distortions or flickering artifacts. More stylization videos are packed in the supplementary material.

**Photorealism.** In photorealistic style transfer, the most important principle is to change color styles of images without resulting distortions or artifacts. Meanwhile, photorealism means that stylization results should looks like taken from cameras. Figure6 compares ColoristaNet with state-of-the-art methods in terms of photorealism. When zooming in to check details of local image patches, no obvious unpleasant artifacts in results produced by ColoristaNet.

**Coherency.** For video stylization, maintaining temporal coherency is vital in many real applications. In figure 5, from left to right are stylization results at different time steps. There are three different style images in large style variations for each video. A Gaussian smoothing function is exploited to smooth the stylization vectors among frames with different style references. Stylized video frames change smoothly without any flickers. Nor there is any structural inconsistency between stylized video frames. This proves that ColoristaNet is able to produce temporally coherent videos even when there are multiple style references. Please refer to our supplementary material for videos generated by ColoristaNet (with single style reference and multiple style reference).

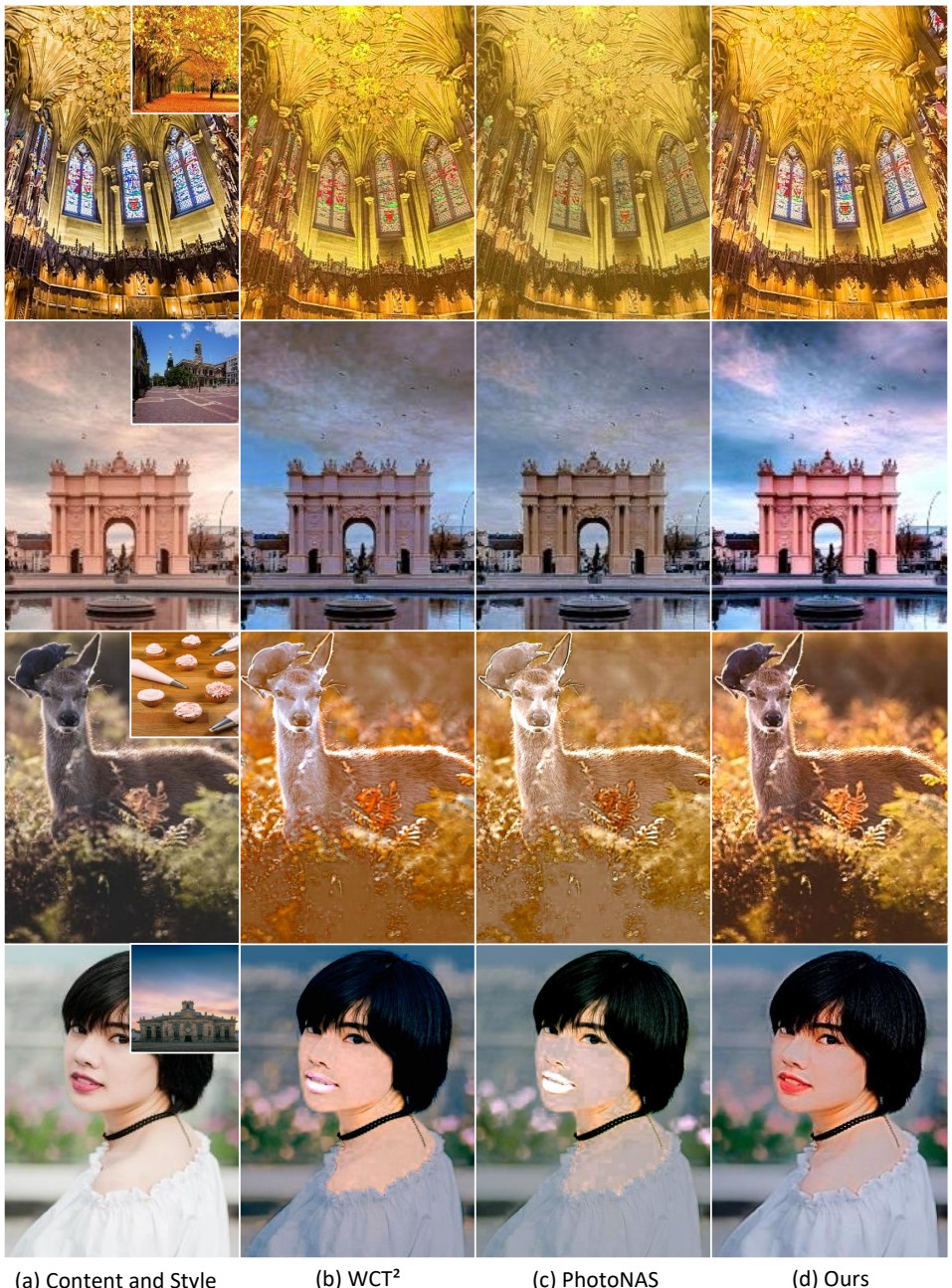

(a) Content and Style     (b) WCT²     (c) PhotoNAS     (d) Ours

Figure 6: Visual comparison with popular algorithms including WCT$^2$ (Yoo et al., 2019), PhotoNAS (An et al., 2020) and our ColoristaNet. Images in the first column and their top right corners are content images and their style counterparts. Each row contains stylization results rendered by different styles. While ColoristaNet generates photorealistic results, other methods either damage image structure or produce over-stylization.

## 4.3 QUANTITATIVE COMPARISON

**Quantitative Metrics.** Following PhotoNAS (An et al., 2020), we evaluate the stylization results with SSIM, LPIPS (Zhang et al., 2018), content loss (Gatys et al., 2016) and Gram loss (Gatys et al., 2016). SSIM, LPIPS (Zhang et al., 2018), content loss (Gatys et al., 2016) are employed to measure the strcture similarity between two images, and gram loss is used to calculate the style distance of two images. We randomly select 100 video/style image pairs to evaluate the performance of state-of-art

algorithms. We use official codes and models provided by PhotoWCT (Li et al., 2018), WCT$^2$ (Yoo et al., 2019) and PhotoNAS (An et al., 2020) to generate stylization results. In Table 1, the proposed ColoristaNet achieves better scores in SSIM, LPIPS, the content loss, and a comparable gram loss. It means our ColoristaNet has a stronger ability in preserving structural details.

Table 1: Quantitative comparison with state-of-art photorealistic style transfer algorithms.Higher SSIM scores mean test images are more similar to input contents with fine details. Lower LPIPS scores mean higher perceptual similarities between stylization results and content images.

| Method | PhotoWCT(full) | WCT$^2$ | PhotoNAS | Ours |
|---|---|---|---|---|
| SSIM↑ | 0.548 | 0.555 | 0.737 | **0.785** |
| LPIPS↓ | 0.464 | 0.391 | 0.326 | **0.223** |
| Content Loss↓ | 11.035 | 7.256 | 4.351 | **2.427** |
| Gram Loss↓ | **0.00025** | 0.00032 | 0.00028 | 0.00026 |

Table 2: User study results.

| Method | PhotoWCT(full) | WCT$^2$ | PhotoNAS | Ours |
|---|---|---|---|---|
| Photorealism | 2.20 | 2.77 | 2.67 | **3.57** |
| Stylization | 2.03 | 2.70 | 2.17 | **3.40** |
| Coherency | 2.00 | 3.37 | 3.27 | **3.70** |
| Overall quality | 2.23 | 3.03 | 2.77 | **3.57** |

Table 3: Computing-time comparison.

| Image Size | PhotoWCT(full) | WCT$^2$ | PhotoNAS | Ours |
|---|---|---|---|---|
| 600×360 | 0.408s | 2.13s | 0.124s | **0.068s** |
| 854×480 | 0.495s | 2.15s | 0.175s | **0.111s** |
| 1280×720 | 0.811s | 4.267s | 0.383s | **0.225s** |
| 1920×1080 | 1.430s | 4.362s | 0.564s | **0.482s** |

**User Study.** We recruited 30 testers who are not connected with this project to evaluate the quality of the stylization results. The testers are asked to take the quality of details in the image and the stylization effects into consideration during their evaluation. Images are rated on a scale of 1-5, where higher scores stand for better stylization results. In total, we collected 900 responses (30 videos × 30 users) for each kind of method. As shown in Table 2, our approach perform best for photorealism, stylization effects, temporal coherency and overall quality.

**Inference Speed.** To demonstrate the efficiency of our method, we compare the inference speed of the different models. We use GeForce RTX 3090 GPU to test all state-of-the-art methods. We randomly select 5 different videos, each of which contains 80 frames, and compute the average running time of each method. Furthermore, we also test the speed of these algorithms in different resolutions. As shown in Table 3, our ColoristaNet is much faster than other methods.

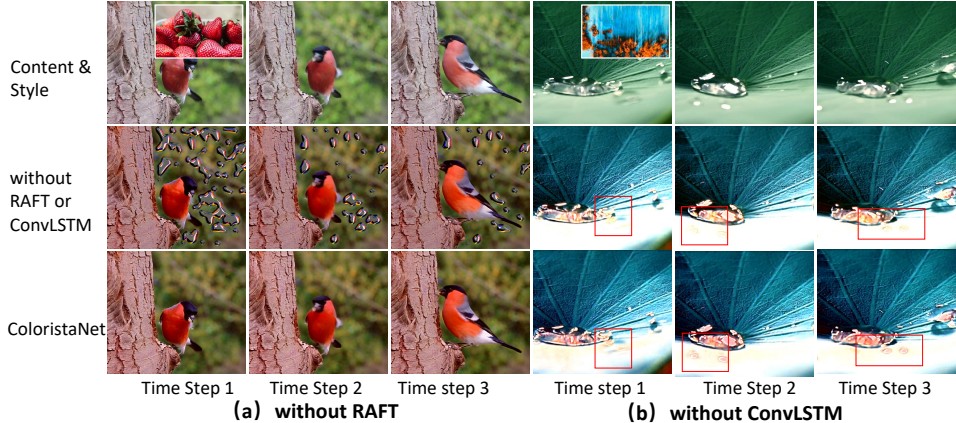

Figure 7: Investigation of the effectiveness of RAFT and ConvLSTM in ColoristaNet.

## 4.4 ABLATION STUDY

**Whether optical flow estimation and ConvLSTM are necessary?** To check whether RAFT together with ConvLSTM are necessary for video style transfer, we remove them to test the performance of ColoristaNet. From Figure7, we can find that when we remove RAFT optical flow estimation,

ColoristaNet will generate noticeable artifacts. When we remove all ConvLSTM units, some images details disappeared. It shows both RAFT and ConvLSTM are necessary.

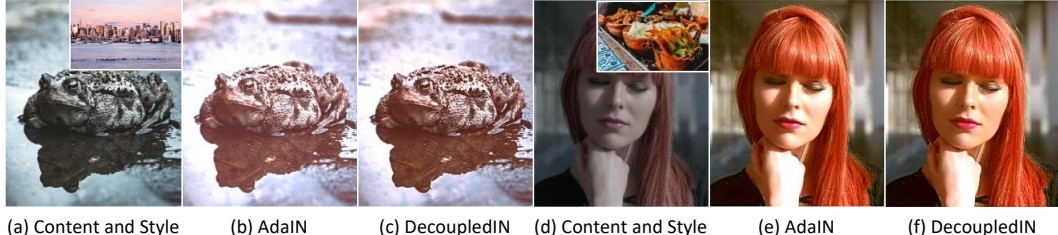

(a) Content and Style     (b) AdaIN     (c) DecoupledIN     (d) Content and Style     (e) AdaIN     (f) DecoupledIN

Figure 8: Visual comparison of the results produced by AdaIN and DecoupledIN.

**Whether DecoupledIN is important to get good results?** To verify decoupledIN's ability in preserving subtle image details, we replace all decoupledIN modules in ColoristaNet with AdaIN. As shown in Figure 8, AdaIN is powerful in generating synthesised images with good stylization effects and keeping photorealism. However, if we zoom in to see more details, we find that AdaIN will overwrite some image details and produce images that seem to be "overexposed".

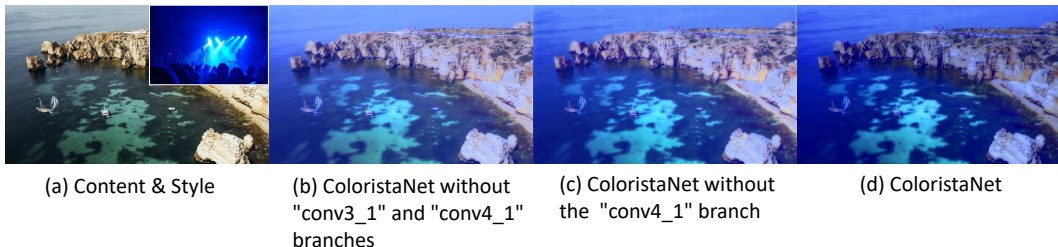

(a) Content & Style     (b) ColoristaNet without "conv3_1" and "conv4_1" branches     (c) ColoristaNet without the "conv4_1" branch     (d) ColoristaNet

Figure 9: Results of ablation on the multi-scale feature fusion scheme of ColoristaNet. (d) is complete multi-scale feature fusion scheme, (c) removes the feature transformation at the "$conv4\_1$" stage, (b) further removes the feature transformation at the "$conv3\_1$" stage on the basis of (c).

**Whether multi-scale features are useful in the decoder?** We remove feature maps produced by "$conv4\_1$" and "$conv3\_1$" stages and compare their results with ColoristaNet in Figure 9. From left to right, we list the stylization results produced by ColoristaNet with two, three, and four different feature scales. These results demonstrate that multi-scale features can generate better stylization results. More results in Appendix B.4 indicates more feature scales will lead to less artifacts.

## 5 CONCLUSIONS, LIMITATIONS AND FUTURE WORKS

In this paper, we propose ColoristaNet, a photorealistic video style transfer network, along with a removal-and-restoration training pipeline. ColoristaNet learns color stylization in a self-supervised manner and generates stylization results looking as if taken from cameras. Two important components of ColoristaNet are decoupled instance normalization and ConvLSTM units that can implement arbitrary style transfer while preserving salient image structure and temporal stylization coherency. Experiments show that results of ColoristaNet have strong visual quality and high artistic value.

**Limitation.** However, since ColoristaNet makes a balance between stylization results and photorealism, it doesn't completely embed styles of reference images into input videos like that of other methods. According to our experiments, in many scenarios, enforcing stylization results to have the exactly same style of references will make images looks like paintings not real photos. Besides, we exploit RAFT to compute optical flow and many other modules with heavy computational cost to conduct style transfer. This makes ColoristaNet unable to run in realtime on a single NVIDIA GeForce RTX 3090 GPU, which deserves thorough analysis in the future. In the future, we will try to work on this problem to alleviate these limitations. Our techniques can be used in many social media platforms and we haven't found obvious negative societal impact of ColoristaNet.

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

# Appendix

## A    IMPLEMENTATION

### A.1    SETTINGS

**Dataset.**    ColoristaNet is trained by videos collected from HMDB (Kuehne et al., 2011) and Youtube VOS (Xu et al., 2018) as content inputs and images collected from Internet as style images. We have around 6000 videos and around 24000 images in our training set. During test, we download videos from Youtube (YouTube), Videvo (VideoNet) and other websites, and select style images that can generate pleasant stylization results.

**Evaluation.**    To check the color stylization ability of ColoristaNet, we conduct style transfer on various high-definition videos with different style images as shown in Figure 5. We compare with photorealistic image style transfer algorithms, such as WCT$^2$ (Yoo et al., 2019), PhotoNAS (An et al., 2020). We directly conduct style transfer frame by frame using their official codes. We can't compare with photorealistic video style transfer algorithms MVStylizer (Li et al., 2020) and Xia et al. (Xia et al., 2021) and other methods, since their source codes are not released. We conduct both quantitative comparison and a user study to evaluate different algorithms.

**Training.**    All experiments are implemented with PyTorch (Paszke et al., 2019), and run on two NVIDIA GeForce RTX 3090 GPUs with 24 GB RAM. Parameters of VGG-19 (Simonyan & Zisserman, 2014) and RAFT (Teed & Deng, 2020) are initialized with pretrained weights and are fixed during training. We use the SGD optimizer with momentum 0.9 and basic learning rate 1e-5 to optimize parameters of ColoristaNet in 80 epochs. We schedule our learning rate a bit different from common practise. The learning rate at the first epoch is set to 0.01, and then is decreased to 1e-5 in the next five epochs. After that we apply cosine decay for the rest epochs. Images are cropped and resized into the resolution $128 \times 128$ during training.

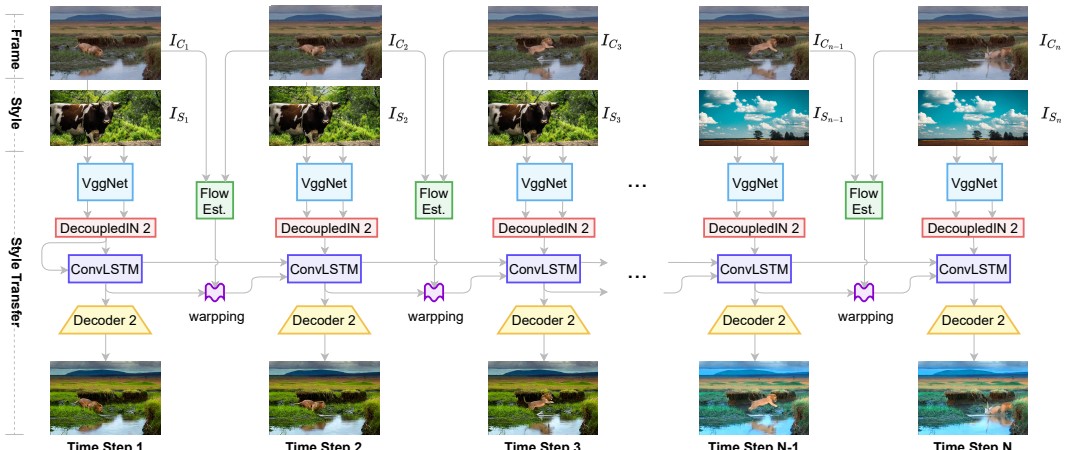

Figure 10: Inference with style transfer network. For a video clip, each frame is paired with a style reference to pass VGG-19 feature extractor, decoupled instance normalization, ConvLSTM and the decoder to generate the final output. There may be multiple style references at the same time. ColoristaNet can conduct arbitrary style on videos of any length according to users' preference.

### A.2    CONFIGURATIONS OF STYLE TRANSFER NETWORK

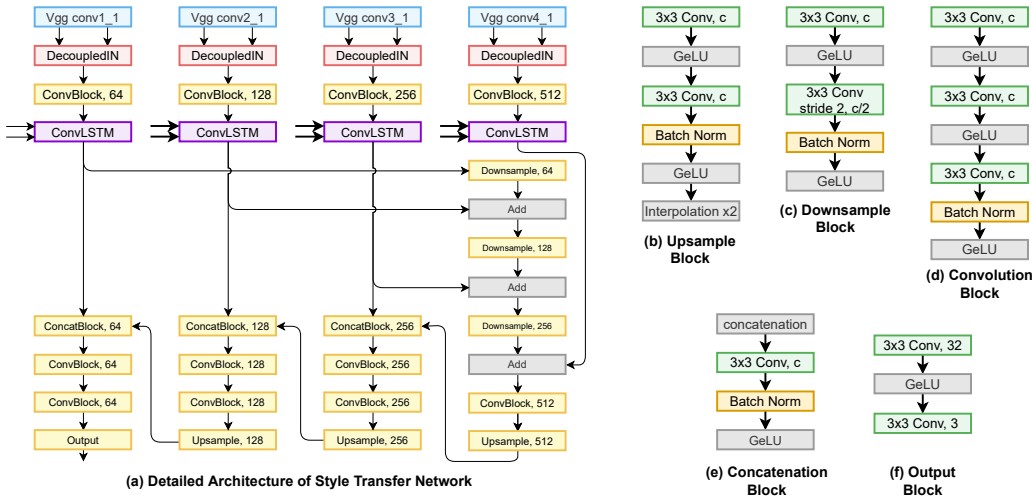

Figure 11: Detailed illustration of style transfer network. There are five shared blocks in a style transfer network: (b) Upsample block, (c) Downsample Block, (d) Convolution Block, (e) Concatenation Block, and (f) Output Block. Structures of style transfer networks in style removal and restoration are slightly different. In style removal network, there are no ConvLSTM (Shi et al., 2015b) units.

Figure 11 gives detailed architectures of ColoristaNet. The style transfer network has a U-net (Ronneberger et al., 2015) style encoder-decoder structure. During training, networks are shared by different time steps. The number of convolution filters in each block is denoted with $c$. "ConvBlock, 64" stands for the filter number in a convolution block is 64 and the kernel size of convolutional layers is $3 \times 3$. Other blocks have the similar notations. The style removal network and style restoration network have a similar architecture, except that there are no ConvLSTM (Shi et al., 2015b) units across style removal networks.

## B COLORISTANET: DETAILS, ADDITIONS AND ABLATIONS

To generate photorealistic stylization results which look like taken from cameras, ColoristaNet exploits a set of training strategies and micro designs to avoid structural distortions and painterly artifacts, including self-supervised learning, decoupled instance normalization, flow estimation network (RAFT) (Teed & Deng, 2020), ConvLSTM (Shi et al., 2015a), multi-scale feature learning and etc. Here, we give more details, additional experiments and ablations on the designs and choices of these different modules.

### B.1 SELF-SUPERVISED LEARNING IN COLORISTANET

Self-supervised learning obtains supervisory signals from unlabeled data itself and thus leverages underlying structure and common representation in data, which has achieved great success in natural language processing (Devlin et al., 2018; Brown et al., 2020) and computer vision (Chen et al., 2020a; Caron et al., 2021; He et al., 2022). In unpaired image-to-image translation, CycleGAN (Zhu et al., 2017a) introduces consecutive image-to-image translations in cycles to ensure content consistencies in images with the help of generative adversarial networks (Goodfellow et al., 2014). In this paper, we exploit the self-supervised learning strategy for photorealistic style transfer. Our motivation here is that if the style of an image can be replaced without hurting subtle structures arbitrarily, its style can be recovered by using itself as the style reference. Such an assumption holds in photorealistic style transfer because the underlying image structure remains unchanged during color style transfer. So in our training pipeline, we apply two ColoristaNets that are responsible for style removal and style restoration respectively. When styles of images are removed, the style transfer or restoration task becomes a fully supervised one. As shown in Figure 12, such a strategy has been proven to be the key of the success of ColorsitaNet.

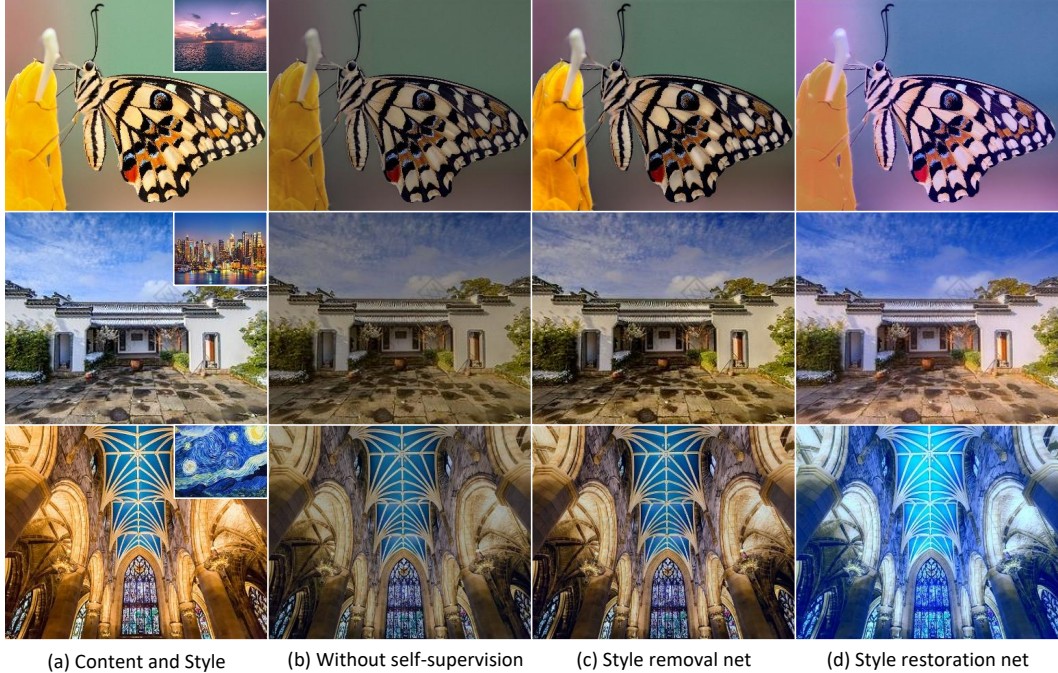

(a) Content and Style  (b) Without self-supervision  (c) Style removal net  (d) Style restoration net

Figure 12: Stylization results comparison of different style transfer networks: (a) Content and style images, (b) A single ColoristaNet simply trained with content loss (without the removal-restoration pipeline), (c) The style removal ColoristaNet in the proposed training pipeline, (d) The style removal ColoristaNet in the proposed training pipeline.

**Style Removal.** During style removal, style transfer network overwrites styles of images with given style references through decoupled instance normalization at different feature resolutions. Conducting style transfer without resulting any distortions or artifacts is the most important principle in style removal. We implement style removal with decoupled instance normalization and give detailed introduction and analysis in Appendix B.2. Meanwhile, to ensure image structures to be full preserved, we simply exploit the content loss (Gatys et al., 2016) to enforce the structure consistency. As is known that style loss (Gatys et al., 2016) can conduct mixed transfers of both texture and color, it will result in painterly distortions of image structures. Figure 12 visualizes stylization results of the style removal network. It can be found that image structures of content images are preserved and original image styles are partly removed. Its stylization effects are unsatisfactory and look just like paintings.

**Style Restoration.** Following the assumption described above, the style restoration network just takes the style removal result in as the content input and use the original image as the style reference to conduct style transfer. Thus, the stylization results can be expected to be the same with the original content image. In this way, the content image, the style reference and the stylization result can be defined clearly. We use the style transfer network in the same architecture with that in the style removal part to conduct supervised style transfer. Again, no style loss is exploited to enforce good stylization. Surprisingly, we can find that such a strategy can help to generate good stylization results without obvious distortions or artifacts. We attribute this to that the linear feature transformations in DecoupledIN and the self-supervised learning framework without style loss. Besides, we find that training the style removal and restoration networks jointly will make the training much more stable and converge faster. To validate the effectiveness of the two stage style transfer framework, we conduct an ablation study to check whether the self-supervised learning strategy is necessary. We train ColoristaNet with a randomly selected style reference to perform parameter learning without self-supervision. Figure 12 shows the stylization results produced by ColoristaNet without self-supervision, the style removal and restoration ColoristaNets respectively. It can be found that without the self-supervised learning strategy, ColoristaNet can not transfer the style of a reference image to the target successfully.

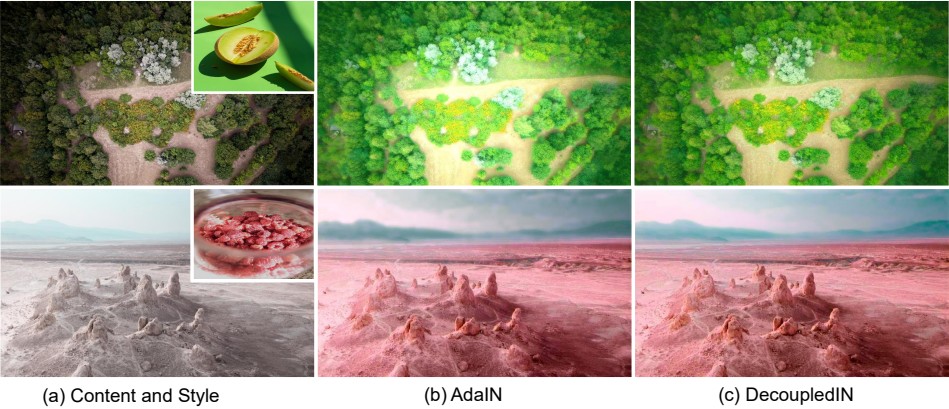

(a) Content and Style          (b) AdaIN          (c) DecoupledIN

Figure 13: Visual comparison of the results produced by AdaIN and DecoupledIN.

## B.2 Decouple Instance Normalization

Transferring styles of images arbitrarily through feature transformations has been widely accepted in style transfer (Huang & Belongie, 2017; Li et al., 2017c; 2018; Yoo et al., 2019). In (Huang & Belongie, 2017), Huang *et al.* proposed adaptive instance normalization (AdaIN) to aligns the mean and variance of the content features with those of the style features to achieve arbitrarily artistic style transfer without iterative optimization process. Li *et al.* (Li et al., 2017c) conducted universal style transfer through the whitening and coloring transforms (WCT) to match feature covariance of the content image to a given style image. PhotoWCT (Li et al., 2018) brought the idea of conducting style transfer through feature transformations from WCT (Li et al., 2017c) to perform photorealistic style transfer. However, the feature transformation in WCT is nonlinear, and thus lead to distortions in image structures obviously. PhotoWCT (Li et al., 2018) exploits a photorealistic smoothing term to ensure local consistency in pixel intensities. But in many cases, there are obvious artifacts produced by WCT that can not be removed by the smoothing term. Besides, the smoothing term also make images blurry and lost some subtle local contrast. Yoo *et al.* (Yoo et al., 2019) proposed a wavelet corrected transfer based on whitening and coloring transforms ($WCT^2$) to avoid introducing additional masks and unfavorable blurry artifacts. Although these methods are powerful and can conduct arbitrary style transfer, they can't achieve photorealistic stylization results. According to our analysis, the feature whitening and coloring transforms in WCT (Li et al., 2017c) can match the feature correlation of content images to that of style references exactly, but it will change the local feature contrast which maintains image structures. AdaIN has very nice properties that won't damage local feature contrast but it can not match the feature statistics of style references compactly and thus lead to inferior stylization effects.

In this paper, we aim to achieve photorealistic stylization while still make synthesized images look like taken from cameras. As shown in Figure 13, AdaIN is powerful to generate synthesized images with good stylization effects and keeping photorealistic. However, if we zoom in to see more details, we find that AdaIN will overwrite some image details and produce images that seem to be "overexposed". We attribute this to that AdaIN aligns feature statistics of content images with their style references in a shared feature space, which may weaken the local structural details. To alleviate this issue and achieve better stylization results, the decoupled instance normalization aims to decompose the feature transformation into a style whitening step and a restylization step. Thus, we need to insert a convolutional layer after the style whitening operation. The motivation of designing the 3x3 convolutional layer with 2c filters is to expand the feature channels to recover the missing information during the whitening step. Then in the subsequent stylization step, we can conduct style transfer in a higher dimension to benefit from the rich information provided by more feature channels. At last, we fuse the transformation with a simple convolution operation. From Figure 13 and Figure 16, we can find that DecoupledIN can preserve more image details and achieve better stylization results.

**Multiple Style Whitening in DecoupledIN.** The structure of DecoupledIN modules with more whitening operations are demonstrated in figure 14 and figure 15. From Figure 16, we can find

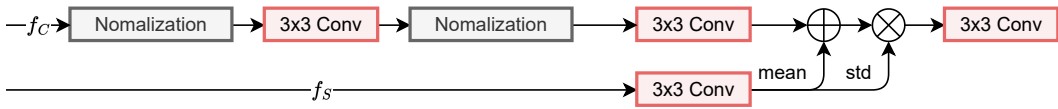

Figure 14: Visualisation of the module structure of DecoupledIN with two style whitening.

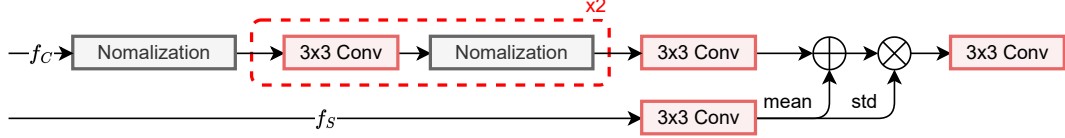

Figure 15: Visualisation of the module structure of DecoupledIN with three style whitening.

| | | | | |
|---|---|---|---|---|
| (a) Content and Style | (b) AdaIN | (c) DecoupledIN | (d) Whitening x 2 | (e) Whitening x 3 |

Figure 16: Visual comparison of AdaIN and DecoupledIN with different style whitening strategies. From left to right: (a)content and style images, (b) AdaIN, (c) DecoupledIN with one style whitening, (d) DecoupledIN with two style whitening, and (e) DecoupledIN with three style whitening. The stylization effects is enhanced from left to right gradually.

that althrough DecoupledIN gets better stylization effects compared with AdaIN, the style of the original content input is remained and combined with the style reference to generate mixed stylization results. To remove the original image style more completely, we conduct multiple style whitening in DecoupledIN and visualaize the results in Figure 16. We conduct the style whitening in DecoupledIN by 1, 2 and 3 times respectively. When looking at these image carefully, we can find that more style whitening will lead to better stylization effects. Since the influence of different feature transformations is hard to be distinguished, please zoom in to check the color variations carefully.

### B.3 FLOW ESTIMATION AND CONVLSTM FOR TEMPORAL CONSISTENCY MAINTENANCE

Optical flow estimation methods (Dosovitskiy et al., 2015; Ilg et al., 2016; Teed & Deng, 2020) and ConvLSTM (Shi et al., 2015a) have been exploited in many video style transfer literature (Chen et al., 2017; 2020b; Anderson et al., 2016; Gao et al., 2020) to ensure temporal coherency. In ColoristaNet, we exploit RAFT (Teed & Deng, 2020) to track pixels across different frames and use ConvLSTM to pass the contextual information to adjacent frames. With RAFT and ConvLSTM, we implement temporal consistent feature transformations and thus can conduct photorealistic video style transfer with high temporal coherency. While as it has been shown in the supplementary videos, simply transferring styles of each frame in videos with image-based photorealistic style transfer algorithms can still generate stylized videos in good quality. It's because that consecutive frames in videos are temporally coherent and we use a Gaussian smoothing function to smooth the style vectors during switches between different style references. However, if we zoom in to check the details of other state-of-the-art methods, such as PhotoWCT (Li et al., 2018), WCT$^2$ (Yoo et al., 2019) and PhotoNAS (An et al., 2020), we can find that there are many structural distortions and flickering artifacts. Since these previous methods can't ensure the photorealism in most of their stylization results, the temporal inconsistency problem are not in the first place to be solved.

**Without RAFT.** To check whether RAFT can maintain temporal consistency during style transfer, we conduct ablation study by removing RAFT from ColoristaNet and still use ConvLSTM to fuse information across different frames. Figure 17 shows that without RAFT, ColoristaNet generates many stylization results with obvious artifacts. It's because without the guidance of optical flow, the information propagation between adjacent frames becomes incorrect. In fact, in some cases, RAFT failed to generate accurate correspondences between image pixels and thus lead to bad stylization effects.

**Without ConvLSTM.** To check whether ConvLSTM can incorporate contextual information among adjacent frames, we remove the ConvLSTM units to test the performance of ColoristaNet. Since ConvLSTM aims to propagate information across different frames, if there is no ConvLSTM, RAFT should be removed too. That means we just conduct video style transfer with image-based algorithms as that of PhotoWCT (Li et al., 2018), WCT$^2$ (Yoo et al., 2019) and PhotoNAS (An et al., 2020). Figure 18 shows two different kinds of scenarios that ConvLSTM is important for ColoristaNet. In the first case, if there is no ConvLSTM module, there will have some flickering artifacts. And in the second case, ColoristaNet with ConvLSTM can ensure style consistency when there are multiple styles to be transferred.

### B.4 EMPLOYMENT OF MULTI-SCALE FEATURES

Since deep convolutional neural networks have multi-scale hierarchical structures and can encode images into features with different semantic levels, it's popular to conduct style transfer with the benefits of multi-scale features (Li et al., 2018; Yoo et al., 2019). PhotoWCT (Li et al., 2018) passed the lost spatial information to the decoder in a hierarchical manner to facilitate reconstructing fine details for photorealistic image synthesis. WCT$^2$ (Yoo et al., 2019) inherited the U-net style structures from PhotoWCT and achieved impressive results. PhotoNAS (An et al., 2020) searched the best architecture for photorealistic style transfer and found that the multi-level stylization strategy is important for style transfer in high quality. We adopt the multi-scale architecture as previous methods to conduct style transfer. We conduct ablations on the multi-scale feature fusion scheme of ColoristaNet. In the style transfer network, there are feature transformations at four different resolutions. Denote the feature transformations in ColoristaNet from lower resolutions to higher resolutions with "$conv1\_1$", "$conv2\_1$", "$conv3\_1$" and "$conv4\_1$" respectively. We remove the feature transformation at the "$conv4\_1$" stage at first, and remove other feature transformations

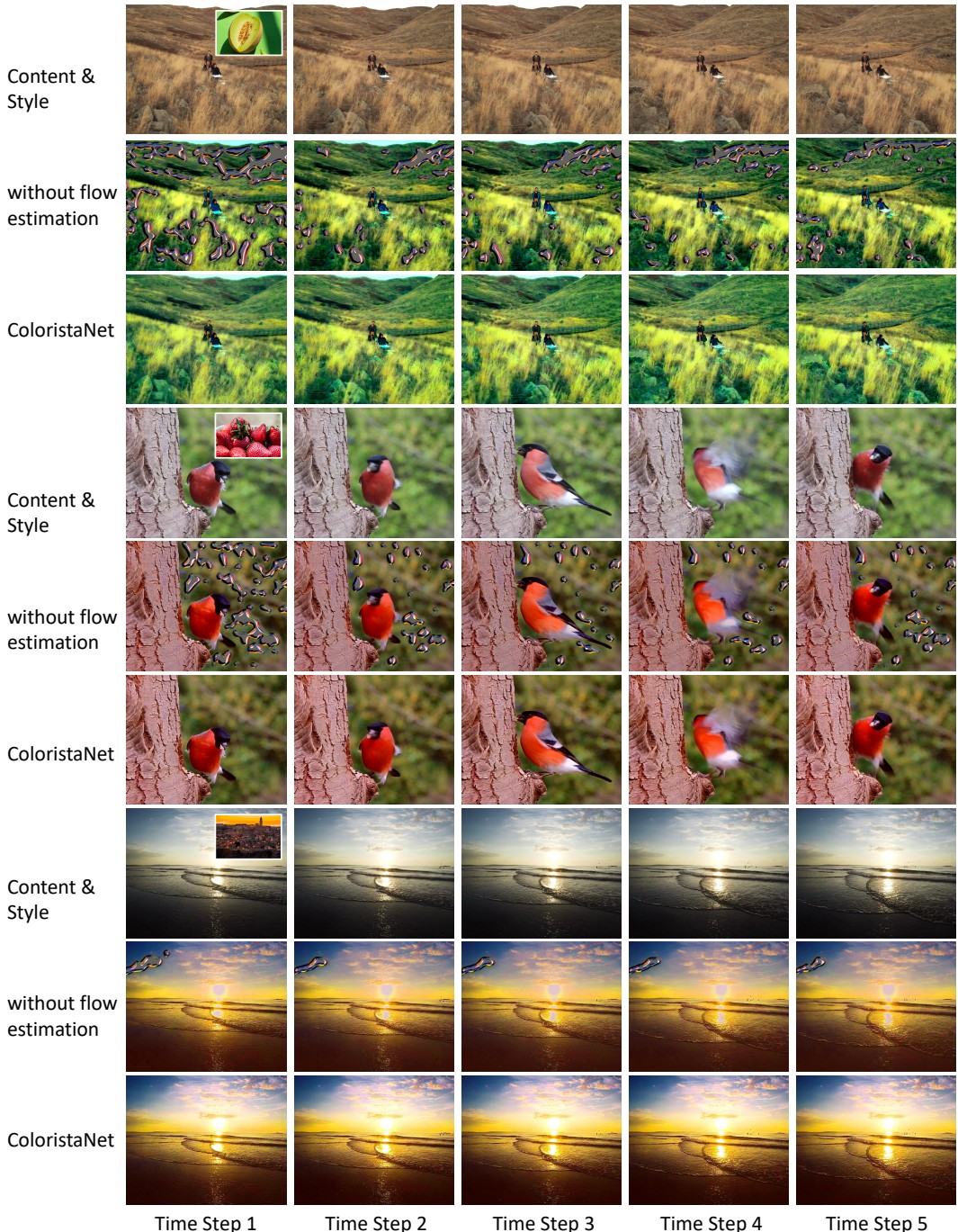

Figure 17: Investigation of the effectiveness of flow estimation network in ColoristaNet. We test ColoristaNet in three different scenarios to see that without flow estimation network (RAFT (Teed & Deng, 2020)), ColoristaNet will generate many artifacts.

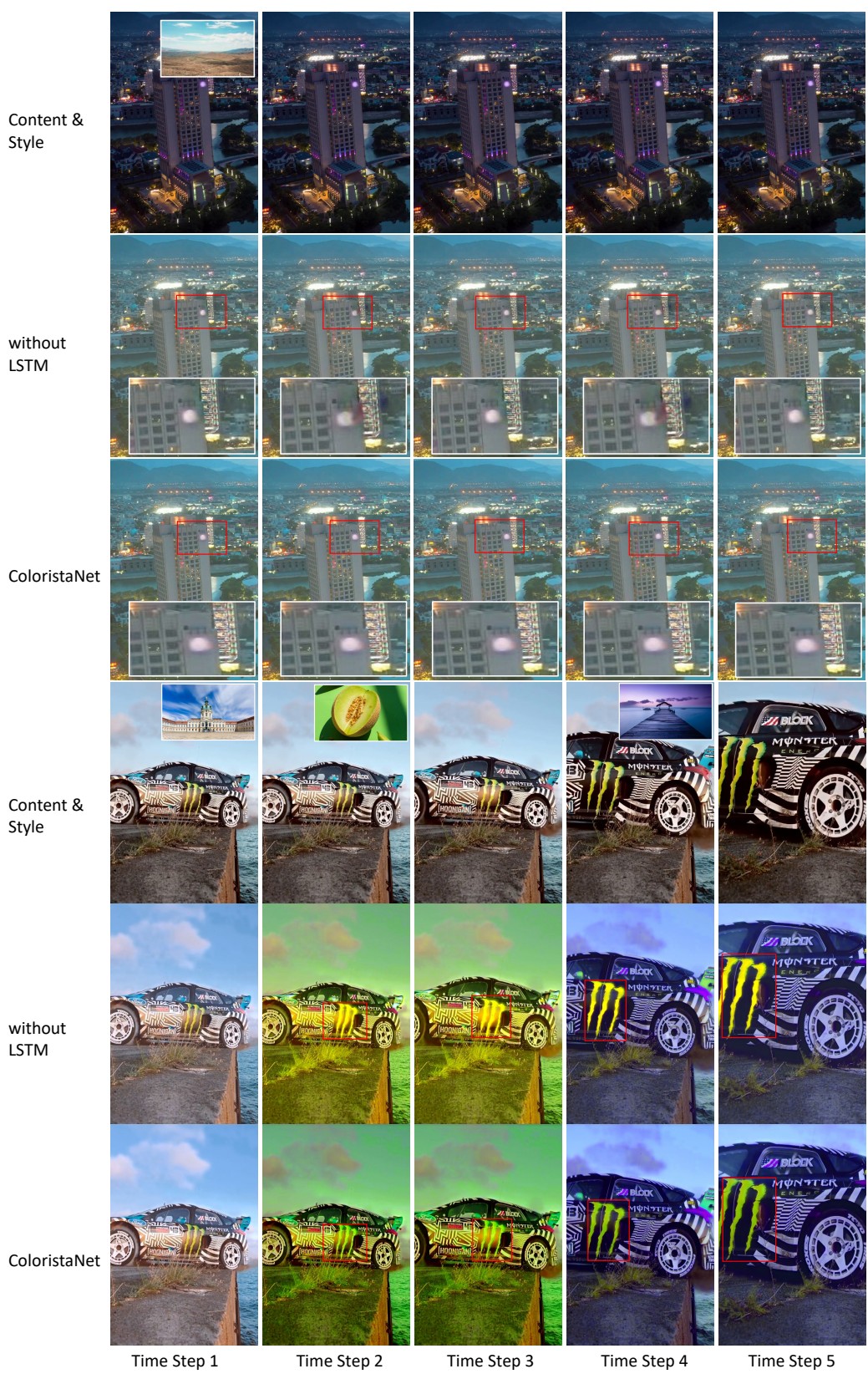

Figure 18: Investigation of the effectiveness of ConvLSTM in ColoristaNet.

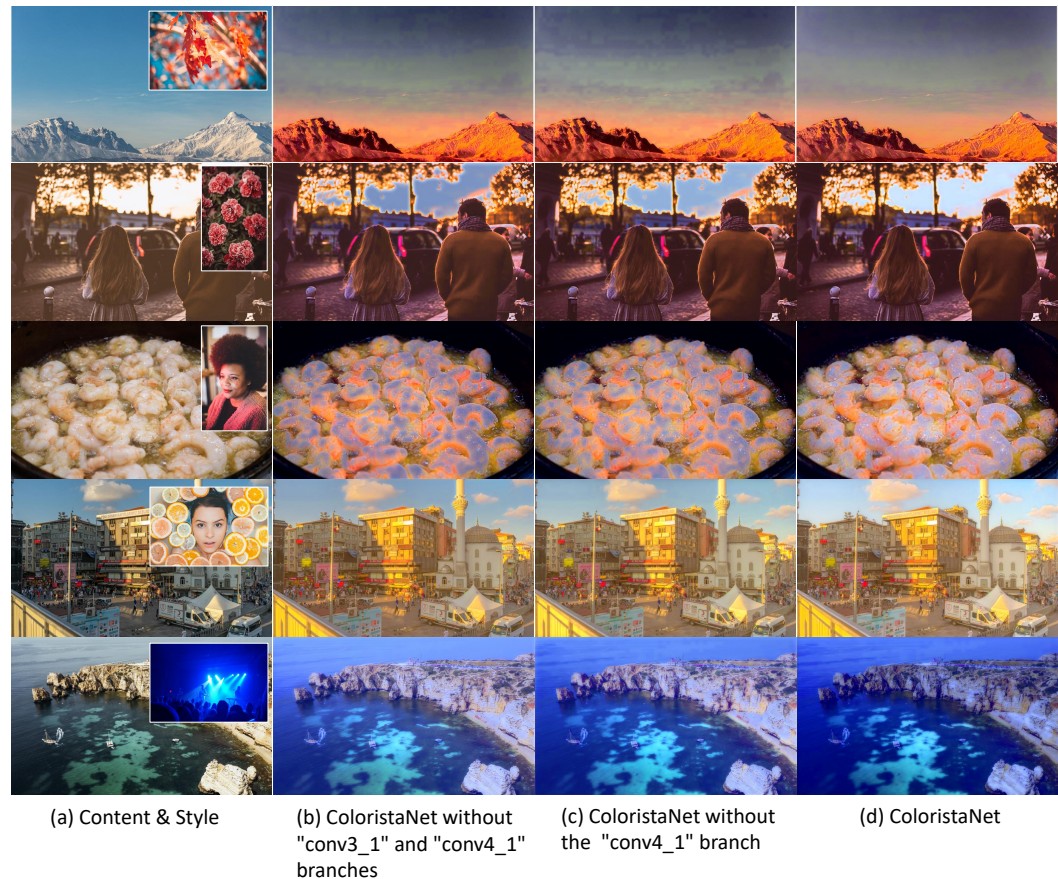

| (a) Content & Style | (b) ColoristaNet without "conv3_1" and "conv4_1" branches | (c) ColoristaNet without the "conv4_1" branch | (d) ColoristaNet |

Figure 19: Results of ablation on the multi-scale feature fusion scheme of ColoristaNet. (d) is complete multi-scale feature fusion scheme, (c) removes the feature transformation at the "$conv4\_1$" stage, (b) further removes the feature transformation at the "$conv3\_1$" stage on the basis of (c).

consecutively to check the stylization results. As shown In Figure 19, from left to right, we list the stylization results produced by ColoristaNet with two, three, and four different feature scales. It can be found that with more different scales, we get better stylization results with less artifacts.

## C   EXPERIMENTAL RESULTS WITH FURTHER ANALYSIS

### C.1   COMPARISON WITH STATE-OF-THE-ART METHODS

Previous state-of-the-art photorealistic style transfer algorithms, such as MVStylizer (Li et al., 2020) and Xia's video style transfer (Xia et al., 2021), don't make their codes publicly available, so we can just compare with image-based photorealistic style transfer algorithms. We compare with four state-of-the-art algorithms including PhotoWCT (Li et al., 2018), WCT$^2$ (Yoo et al., 2019) and PhotoNAS (An et al., 2020). We simply conduct style transfer on image frames independently and put the stylization results together to generate stylized videos. To avoid drastic style change when there are multiple style references, a Gaussian smoothing function is exploited to smooth the stylization vectors among frames with different style references. The Gaussian kernel size is 20. That means for every 20 video frames, their corresponding stylization vectors are smoothed. Figure 20, Figure 21, Figure 27, Figure 28 and Figure 29 give the visual comparison with state-of-the-art algorithms. All previous state-of-the-art algorithms will produce observable blur, structural distortions and flickering artifacts. While ColoristaNet generates videos that look like taken from cameras without any blur, distortions and artifacts. Although ColoristaNet does not transfer the colors of style references to the targets completely, the stylization results are still very competitive.

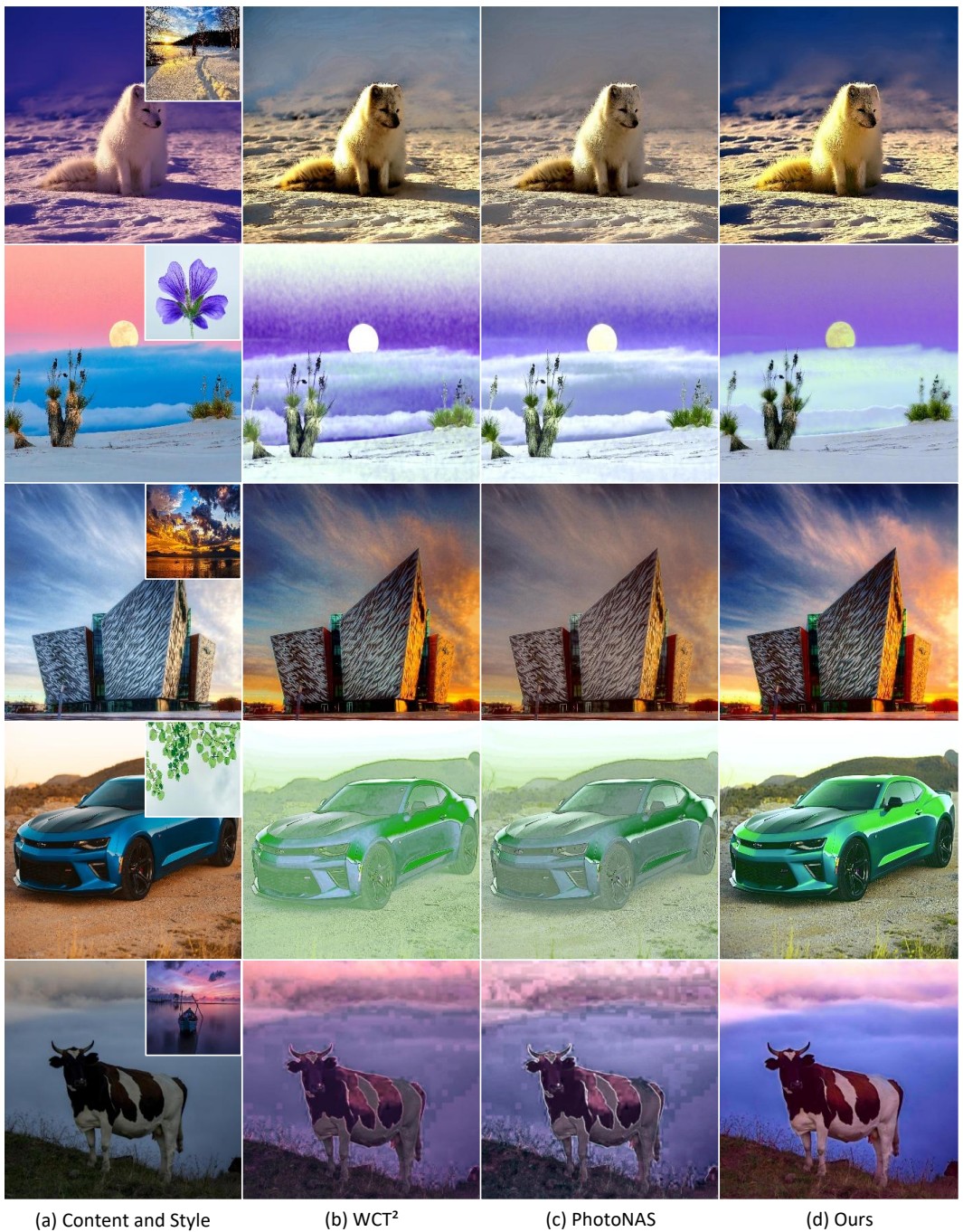

|(a) Content and Style|(b) WCT²|(c) PhotoNAS|(d) Ours|

Figure 20: Visual comparison with popular algorithms including WCT² (Yoo et al., 2019), PhotoWCT (Li et al., 2018), PhotoNAS (An et al., 2020) and our ColoristaNet.

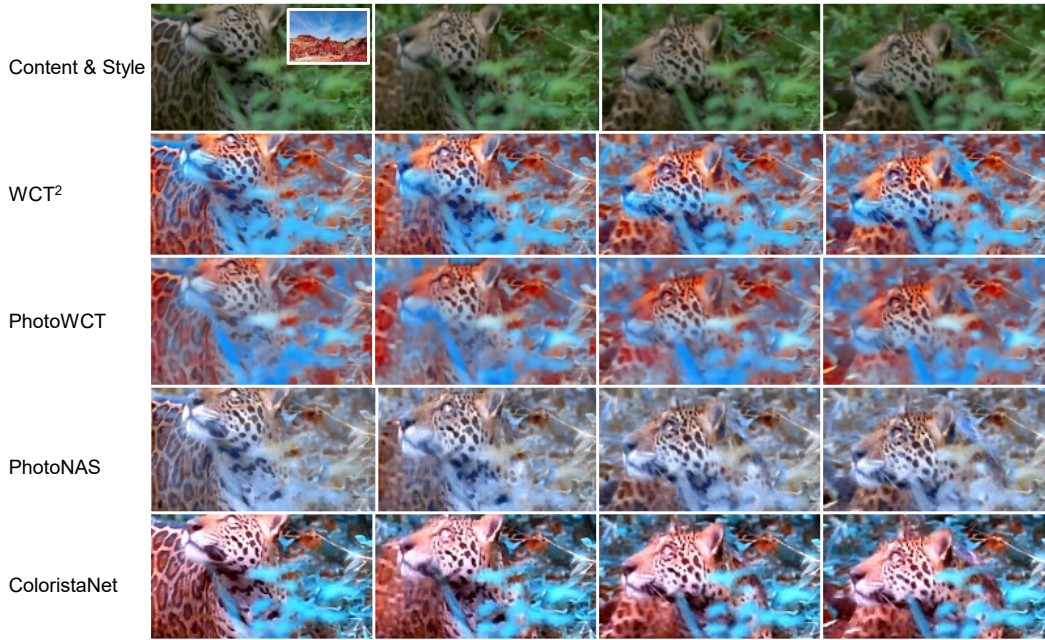

Figure 21: Additional Visual comparison with popular algorithms including WCT$^2$ (Yoo et al., 2019), PhotoWCT (Li et al., 2018), PhotoNAS (An et al., 2020) and our ColoristaNet.

**Contradictions between Stylization and Photorealism.** Separating styles of images from their contents is very difficult since there is no concrete definition of the content and style an image. In Gatys' paper (Gatys et al., 2016), they stated features in higher layer of deep convolutional neural networks capture the high-levl contents interms of objects and their arrangements and can be thought as the content representation. However, as stated in Lapstyle (Li et al., 2017a), the low-level features of the content image is substituted by that of style references and thus will generate unpleasing artifacts. Photorealistic style transfer is some kind of color transfer that need to get the color styles of references image while keeps image content unchanged. In fact, it's hard to achieve a balance between stylization and photorealism. If one want to get a stylization result that matches the style of another image exactly, the color style and the local textures need to be modified. Modifications in local textures will lead to painterly artifacts easily as shown in Figure 26.

### C.2 COMPUTATIONAL COST ANALYSIS

In Table 4, we list the computational cost of different sections of ColoristaNet. There are three main components in ColoristaNet, including VGG-19 feature backbone (Simonyan & Zisserman, 2014), RAFT (Teed & Deng, 2020) and the decoder. There are 3.5M, 1M and 72.2M parameters in them respectively. The VGG-19 feature backbone has only 3.5M parameters because we only use feature maps at the first four convolutional stages. There are relative less parameters in these convolutional layers. RAFT has 1M parameters and the inference speed is very slow compared with other components. The decoder has 72.2M parameters because there four sets of ConvLSTM units, DecoupledIN modules and many other convolutional operations. It can be found that RAFT takes the most time in ColorsitaNet. But when there is no RAFT, ColoristaNet will generate obvious artifacts in many cases as shown in Figure 17. So how to replace RAFT with some other faster methods that can track pixels across different frames is an important problem for future study.

### C.3 STYLIZATION EFFECTS CONTROL

ColoristaNet achieves a balance between good stylization effects and photorealism. We try to improve and control the stylization effects by incorporating a style loss during training, adding a

| Modules | Parameters (M) | Inference Speed (ms) |
|---|---|---|
| VGG-19 backbone | 3.5 | 5 |
| RAFT | 1.0 | 209 |
| the decoder | 72.2 | 14 |

Table 4: Comparison of efficiency between the VGG-19 feature backbone, RAFT and the decoder.

stylization factor in DecoupledIN, and exploiting several DecoupledIN consecutively. We visualize the stylization results in Figure 22 and give some analysis in the following subsections.

**Incorporation of the Style Loss.** We add a stylization loss in both style removal and restoration networks. We have try different weights for the style loss, but these training don't converge.

**Stylization Factor.** Since DecoupledIN can conduct style transfer by matching feature statistics of content images to that of style references, We add a stylization factor $\lambda \in [0, 1]$ in the matching process as follows

$$\text{Whitening}: f'_{C_t,i} = \frac{f_{C_t,i} - \mu\left(f_{C_t,i}\right)}{\sigma\left(f_{C_t,i}\right)},$$

$$\text{Transform}: f''_{C_t,i}, f''_{S_t,i} = \text{Conv}\left(f'_{C_t,i}\right), \text{Conv}\left(f_{S_t,i}\right),$$

$$\text{Reweighting}: \text{Var}_{new}, \text{Mean}_{new} = \lambda\sigma\left(f''_{S_t,i}\right) + (1-\lambda)\sigma\left(f''_{C_t,i}\right), \lambda\mu\left(f''_{S_t,i}\right) + (1-\lambda)\mu\left(f''_{C_t,i}\right),$$

$$\text{Stylization}: g_{t,i} = \text{Var}_{new}\left(\frac{f''_{C_t,i} - \mu\left(f''_{C_t,i}\right)}{\sigma\left(f''_{C_t,i}\right)}\right) + \text{Mean}_{new},$$

$$(3)$$

where $\mu$ and $\sigma$ calculate the mean and standard deviation for each feature channel respectively, and $\text{Var}_{new}$ and $\text{Mean}_{new}$ are the reweighted mean and standard deviation vectors. During the training, we add the stylization factor in the style restoration network, and get the $\lambda$ value randomly using the sampling strategy based on a uniform distribution. During the inference, we set $\lambda$ from 0.2 to 1.0. Figure 22 visualizes the stylization results. It can be found that ColoristaNet can control the stylization effects explicitly.

**Multiple Decoupled Instance Normalization.** In some cases, the style of the original content input is still contained in stylization results produced by ColoristaNet. Here we aim to get stronger stylization results by matching the style of reference images much more closely through adding more DecoupledIN modules consecutively. Figure 23 indicates that when we apply more DecoupledIN modules consecutively in ColoristaNet, the stylization effects will become stronger.

## D    FAILURE CASES AND FURTHER DISCUSSIONS

To examine the robustness of ColoristaNet, we conduct experiments on a plenty of videos to find the failure cases of ColoristaNet. After huge amount of experiments, we find some failure cases and summarize them into three categories, including (1) temporal inconsistency in small regions; (2) under-stylization; (3) over-stylization (caused by extra whitening steps). As in figure 24, we can see that in the original frames, the background color in the red boxes remain unchanged, while in the stylized frames, the coloring is shifting between bright green and dark green. Such inconsistency is caused by properties of DecoupledIN. As the foreground object moves dramatically, the mean and standard deviation of content feature channels will change, which will influence stylizations of both the moving object and the remaining stationary pixels. As such inconsistency can be fixed by ConvLSTM combined with flow estimation in most cases, they become identifiable when flow estimation fails. Among several hundreds of test videos we have examined, we only spotted subtle temporal inconsistencies like Figure 24. As discussed in Appendix B.3, we attribute this to that the input content videos are temporally coherent so in most cases the temporal inconsistency is not obvious. Compared with state-of-the-art methods, our methods sometimes suffer under-stylization (in Figure 25). This is expected since we prioritize photorealism. In the future, we aim to enhance the stylization performance of our model while still maintaining a high level of photorealism. When we

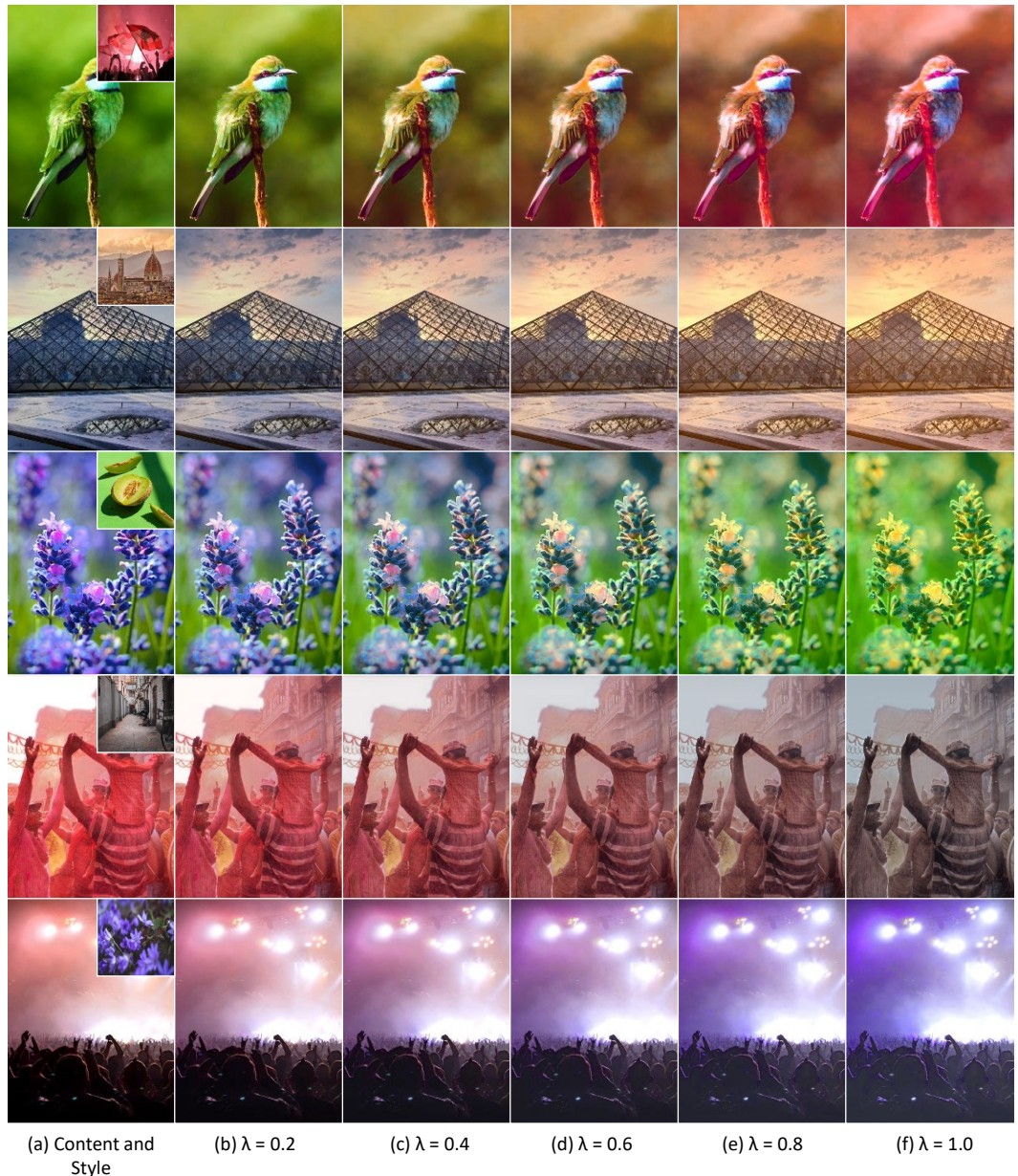

(a) Content and Style  (b) λ = 0.2  (c) λ = 0.4  (d) λ = 0.6  (e) λ = 0.8  (f) λ = 1.0

Figure 22: Illustration of the gradual change in stylisation effect when different style control factors λ are taken. ColoristaNet can control the stylization effect through the stylization factor explicitly.

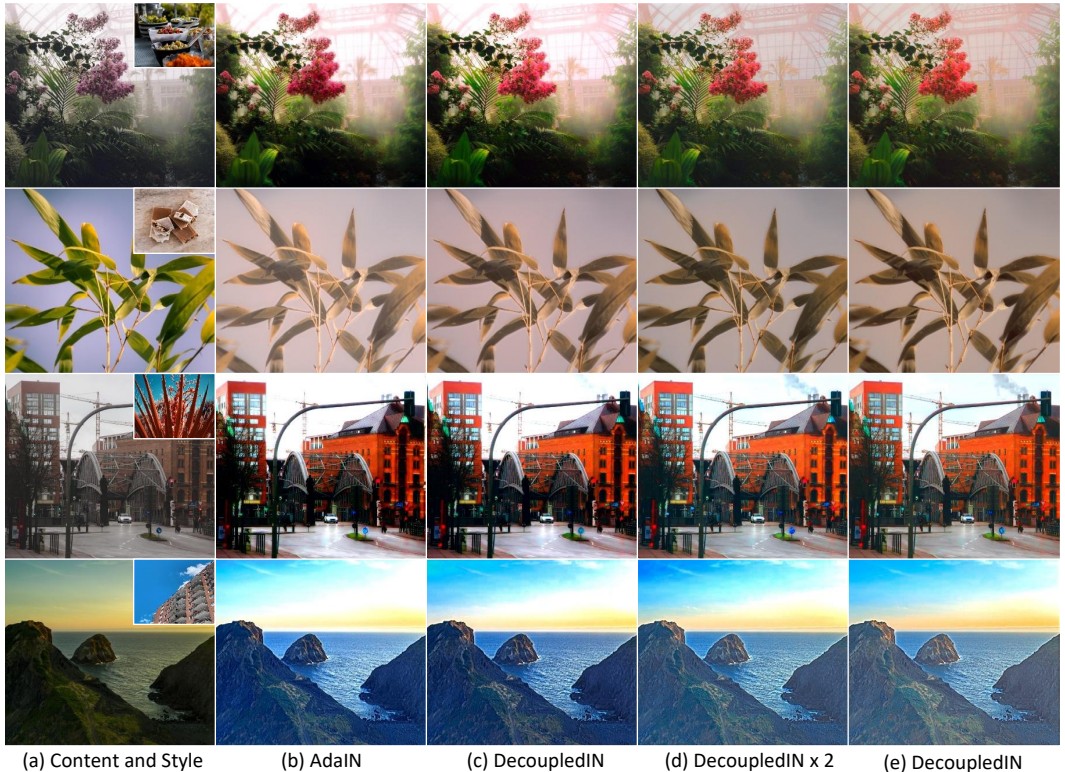

(a) Content and Style     (b) AdaIN     (c) DecoupledIN     (d) DecoupledIN x 2     (e) DecoupledIN

Figure 23: Visualising the results of using AdaIN, decoupledIN, two consecutive decoupledIN and three consecutive decoupledIN.

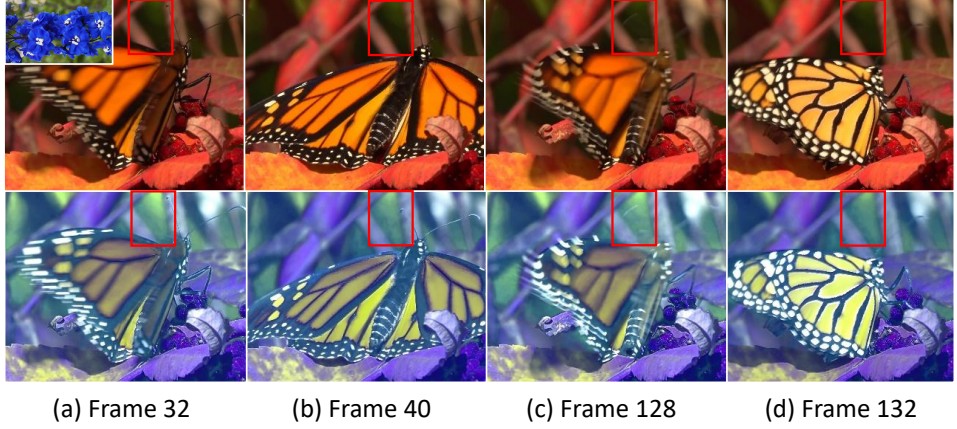

(a) Frame 32     (b) Frame 40     (c) Frame 128     (d) Frame 132

Figure 24: Illustration of failure cases produced by ColoristaNet: Temporal Inconsistency.

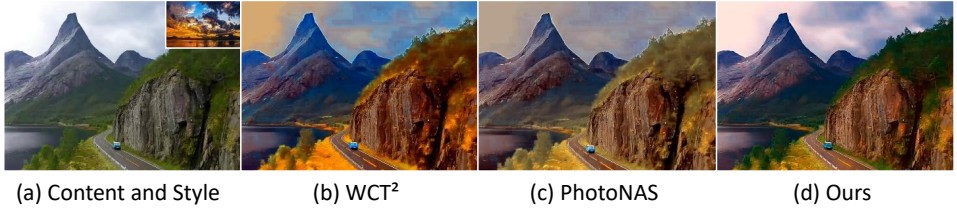

(a) Content and Style     (b) WCT²     (c) PhotoNAS     (d) Ours

Figure 25: Illustration of failure cases produced by ColoristaNet: Under-stylization.

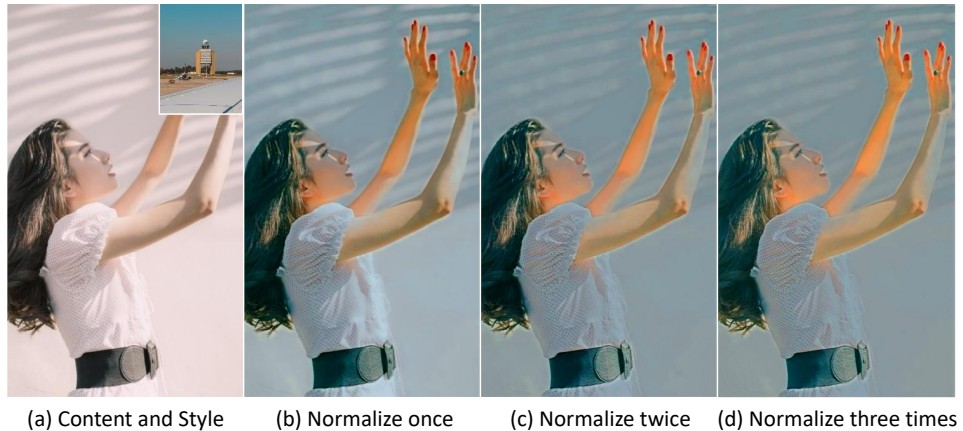

(a) Content and Style    (b) Normalize once    (c) Normalize twice    (d) Normalize three times

Figure 26: Illustration of failure cases produced by ColoristaNet: Over-stylization.

apply multiple whitening steps in our DecoupledIN module, it is possible that too much information is erased, which makes the stylized video look misty (in Figure 26). This is a trade off for stronger stylization effects.

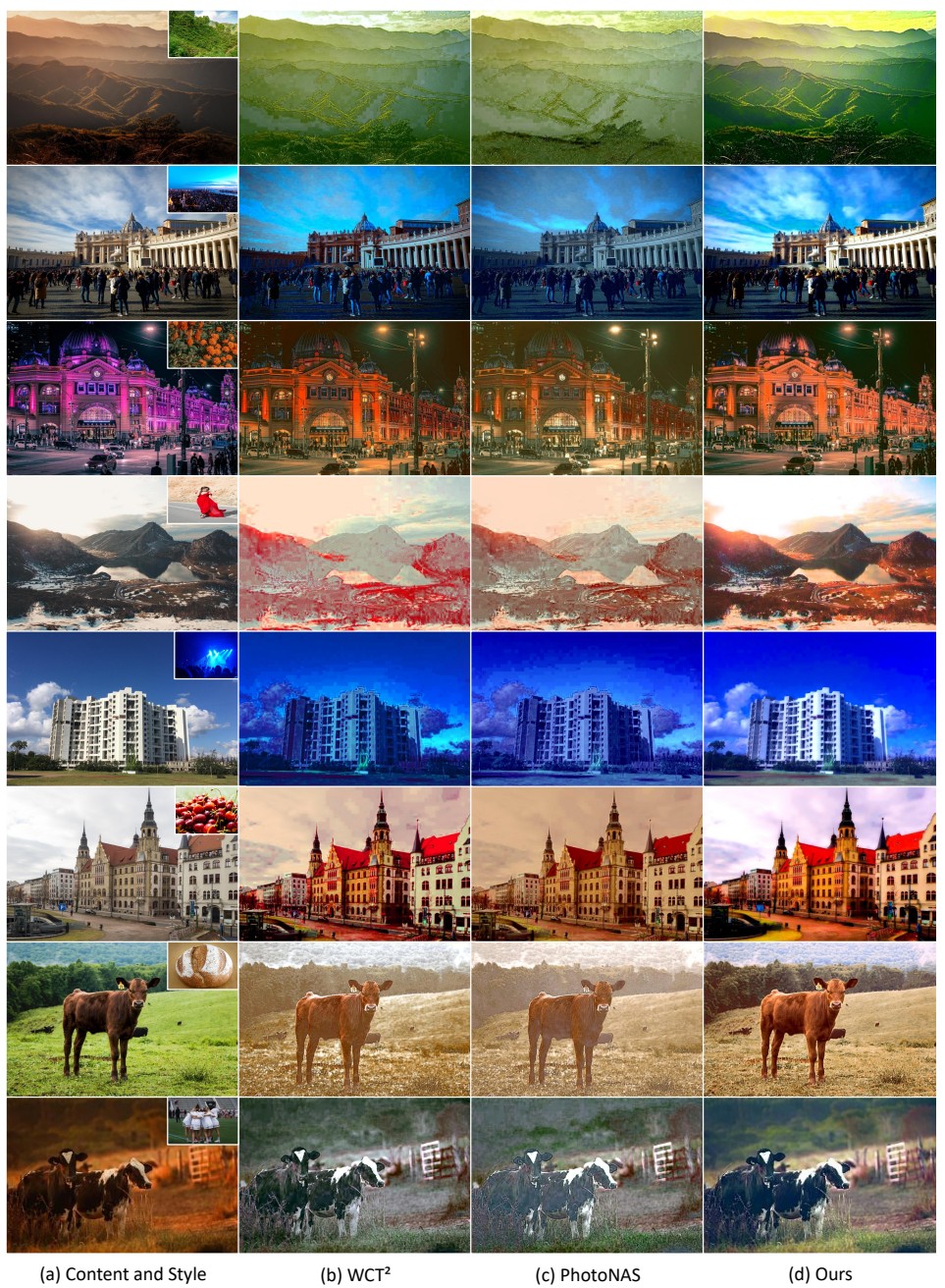

(a) Content and Style      (b) WCT²      (c) PhotoNAS      (d) Ours

Figure 27: Additional comparison between ColoristaNet and state-of-the-art methods.

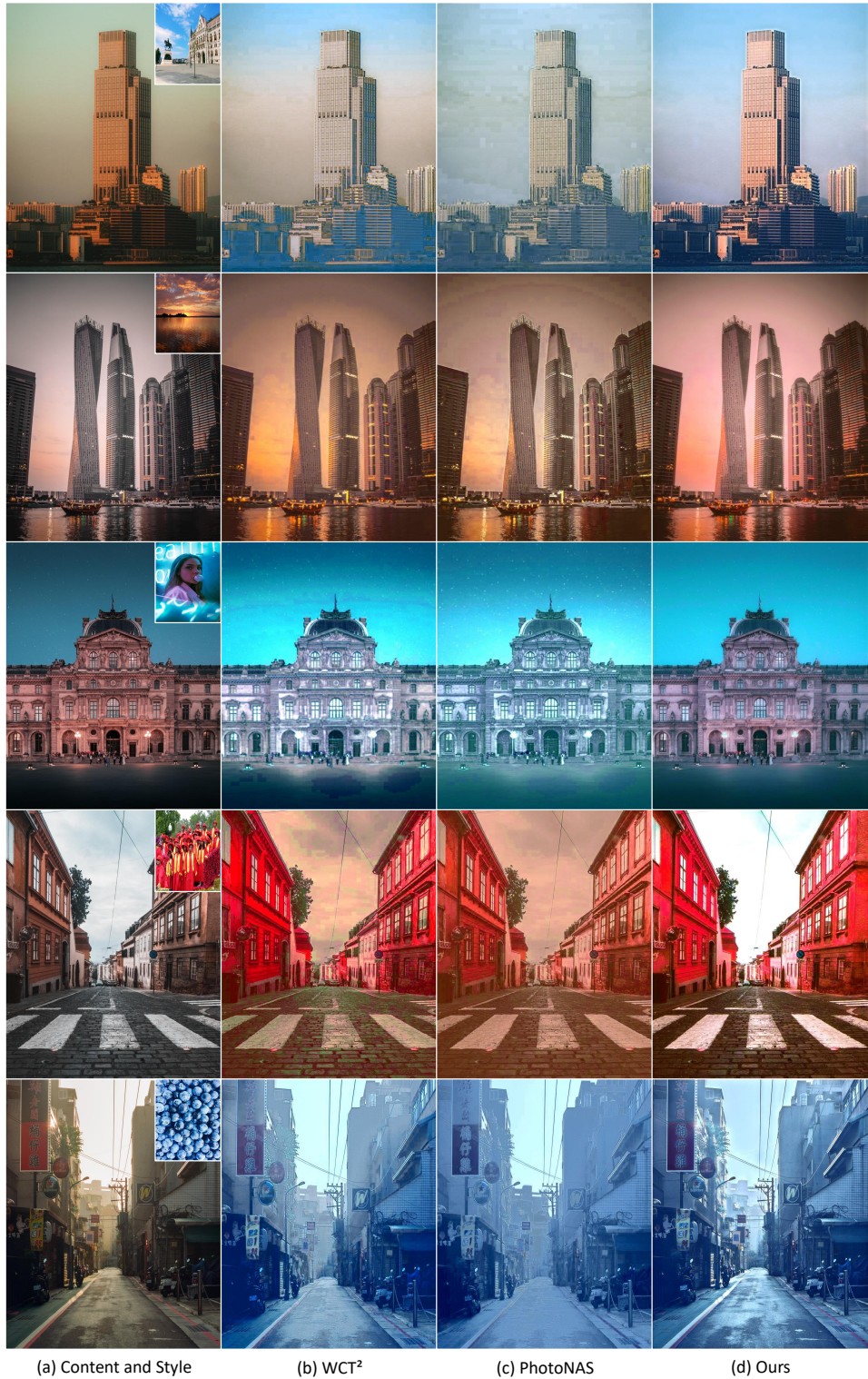

(a) Content and Style     (b) WCT²     (c) PhotoNAS     (d) Ours

Figure 28: Additional comparison between ColoristaNet and state-of-the-art methods.

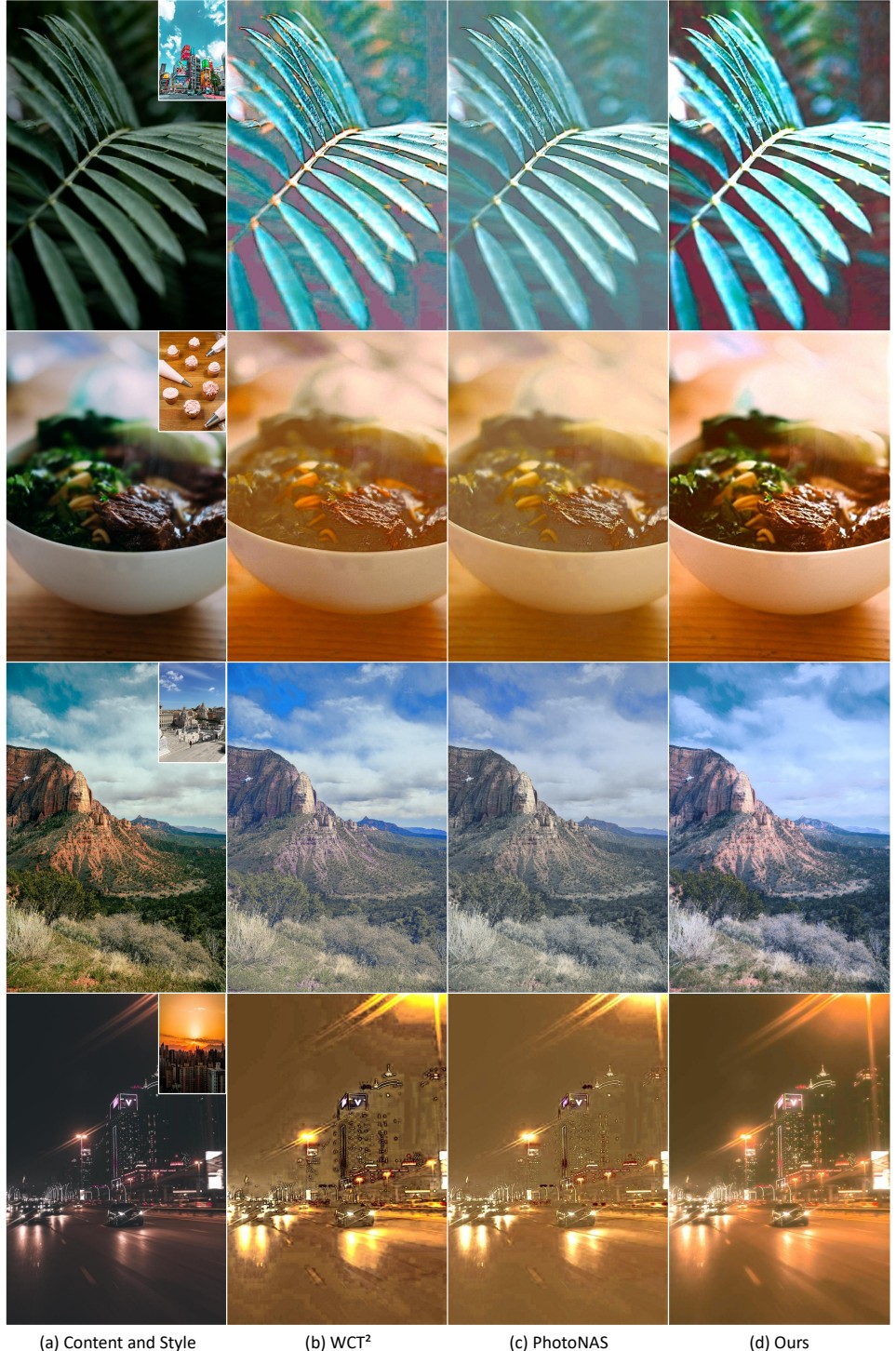

|  |  |  |  |
|---|---|---|---|
| (a) Content and Style | (b) WCT² | (c) PhotoNAS | (d) Ours |

Figure 29: Additional comparison between ColoristaNet and state-of-the-art methods.

