# OpenReview forum: "ColoristaNet for Photorealistic Video Style Transfer"
_ICLR.cc/2023/Conference — Submitted to ICLR 2023_

### Official Review · Reviewer_fibA · 2022-10-19

**Confidence:** 4
**Correctness:** 3
**Technical Novelty And Significance:** 3
**Empirical Novelty And Significance:** 2
**Recommendation:** 3

**Clarity, Quality, Novelty And Reproducibility:**

This paper is clearly written and the main idea is easy to understand. The reproducibility is not hard if given all the network details. The originality of the work is limited considering the ineffectiveness of decoupled AdaIN.

**Strength And Weaknesses:**

**Strengths:**
+ The idea of transforming a style transfer task to a style restoration task is interesting to me. This makes the unsupervised task becomes a self-supervised task, which makes the output’s structure less distorted.

**Weaknesses:**
+ I think the choice of the baseline methods for comparison is not very appropriate. The baselines in this paper (photoWCT, WCT2 and photoNAS) are focusing on improve the photorealism and (or) speed of the style tranfer method of WCT. WCT is known to have strong style effects and high computational cost. Therefore, the starting point of the baselines are to maintain the strong style effects while eliminate structure distortion. But this paper, to me, is more like a pure color transfer method (filter style transfer), and is based on a modified AdaIN, which is known to have less distortion and fast speed. Therefore, in speed comparison in Table 1, the proposed method achieves the fast speed since other baselines all use WCT operation. I recommend some baselines [1-3] that better match the topic of this paper (In [3], the backbone CNNMRF can be replaced with AdaIN).
+ Another problem is the contribution of decoupled AdaIN. First, in Eq. (1), the whitening is to make the feature maps uncorrelated. However, this paper seems only to normalize the feature maps channel-wisely as in AdaIN rather than adjusting the inter-channel covariance. Therefore, why it is called whitening? Second, the decoupled AdaIN seems only add two conv before the addition and multiplication. The novelty is limited. From the visual results, I can hardly tell the difference between AdaIN and the proposed decoupled AdaIN in Fig .4, Fig. 8 and Fig. 13, which weakens the second claimed contribution.
+ For the video part, the optical flow and LSTM are also common components in video style transfer.

**Some small issues:**
+ ’’conv1_1’’ could be revised to ``conv1_1''
+ Fig. 6 is referenced before Fig. 5

[1] 2020 ECCV Filter Style Transfer between Photos

[2] 2020 ECCV Joint Bilateral Learning for Real-time Universal Photorealistic Style Transfer

[3] 2017 BMVC Photorealistic Style Transfer with Screened Poisson Equation


**Summary Of The Paper:**

This paper investigates video color style transfer problem, and proposes a self-supervised style transfer framework. The framework first stylizes the image (style removal) and then reconstruct the original image with the stylized image’s content features and the original image’s style features (style restoration). By doing so, the structure can be better preserved. This paper also proposes decoupled AdaIN for feature fusion and use ConvLSTM and optical flow for temporal consistency.

**Summary Of The Review:**

This paper proposes an interesting idea of style restoration framework for style transfer, which improve the structure preservation. However, the baselines this paper compare do not well matches the topic of this paper. More appropriate baselines could be compared to make the evaluation fairer and more convincing. And the proposed decoupled AdaIN is less novel.

---

### Official Review · Reviewer_K1SY · 2022-10-24

**Confidence:** 4
**Correctness:** 4
**Technical Novelty And Significance:** 2
**Empirical Novelty And Significance:** 2
**Recommendation:** 5

**Clarity, Quality, Novelty And Reproducibility:**

Overall, the clarity is well but there are some problems in writing.
The reproducibility is well. it conduct extensive experiments and show a lot of experimental results to validate the effectiveness.
However, as described in weakness, the novelty is not enough, and it lacks the comparison with the latest work.


**Strength And Weaknesses:**

Strength:
1. This work notice that a lot of previous algorithms will cause unpleasant artifacts or unrealistic results. And it propose a framework which remove style for content image at first and then restore style with the guidance of given style image.
2. This paper conduct extensive experiments, and the visual resutls looks great.

Weakness:
1. From my point of view, almost all components in this work are exsiting models or algorithms proposed by previous work, which makes the novelty of this paper weak.
2. The pipeline in Figure.3 looks complex and ugly. In fact, you can only draw two time step, the repeated others are meaningless for the illusration.
3. the decoupled instantce normalization seems too simple as a main contribution.
4. The paper compares their results with works before 2020, but there are a lot of works[1,2,3] for image style transfering in 2021 and 2022. For fair comparison, I think the results compared with latest methods should be given.
5. There are some problems in writting of this paper, such as 'docouple' in abstract.

[1] Deng, Yingying, et al. "StyTr2: Image Style Transfer with Transformers." Proceedings of the IEEE/CVF Conference on Computer Vision and Pattern Recognition. 2022.
[2] Kim, Sunwoo, Soohyun Kim, and Seungryong Kim. "Deep translation prior: Test-time training for photorealistic style transfer." Proceedings of the AAAI Conference on Artificial Intelligence. Vol. 36. No. 1. 2022.
[3] Wang, Pei, Yijun Li, and Nuno Vasconcelos. "Rethinking and improving the robustness of image style transfer." Proceedings of the IEEE/CVF Conference on Computer Vision and Pattern Recognition. 2021.

**Summary Of The Paper:**

This paper thinks that the summary statistics matching scheme in existing algorithms leads to unrealistic stylization and it proposes a self-supervised style transfer framework to keep photorealism. This frame work contains a style removal part and style restoration part. Besides, it proposes decoupled instance normalization to decompose feature transformation into style whitening and restylization. And it conducts extensive experiments to validate the stylization effecrtiveness of the proposed algorithm.

**Summary Of The Review:**

This work proposes a self-supervised style transfer framework to keep photorealism and decoupled instance normalization to decompose feature transformation into style whitening and restylization. Overall, the paper conducts a lot of experiments to research, which is great. But the designs of the framework and the algorithms lack novelty. And it also lacks the fair comparison with new state-of-the-art methods.

---

### Official Review · Reviewer_AmBd · 2022-10-26

**Confidence:** 4
**Correctness:** 3
**Technical Novelty And Significance:** 2
**Empirical Novelty And Significance:** 2
**Recommendation:** 6

**Clarity, Quality, Novelty And Reproducibility:**

The paper is well written, clear and easy to follow. I would suggest the authors to release the reference implementation if accepted to faciliate reproducibility.

Some limitations:
As the authors pointed out, current framework relies on FlowNet to compute optical flow which is computationally expensive. Along with some other modules, the inference speed prevents ColoristaNet to be a real-time style transfer approach. Would be interesting to explore how the runtime performance can be improved, possibly revisiting some of the modules.

Another limitation, as mentioned above, is that there seems to have no stylization strength control. This is understable since the method aims for photorealistic style transfer and needs to avoid painterly distortions. That said, it would be interesting to explore how to extend the existiing pipeline to support different stylization levels, while maintaining the photorealism and image structure of the scene content.

**Strength And Weaknesses:**

Strengths:

The paper is well written and the exposition is clear.
- Inspired by AdaIN that applied adaptive affine transformations, the authors propose decoupled instance normalization (DecoupledIN) that uses linear transforms in both feature whitening and stylization, capable of conducting photorealistic style transfer effectively.
- To maintain temporal consistency and avoid flickering artifacts, a novel architecture employing optical flow estimation and contextual information with ConvLSTM units is introduced to capture temporal dependencies.
- Sufficient amount of qualitative and quantitative comparisons with prior work and ablation studies are conducted to show the efficiency and quality improvement of the proposed photorealistic style transfer framework.

Weaknesses:

- It seems there is no explicit control, i.e., hyperparameters, to balance the content and style in the stylization output. It is known that more stylized results typically tend to present painterly distortions instead of maintaining the salient image structure. Still, it would be nice to present such extensions or simply show a figure to display results with different level of stylization.
- Show some failure cases. For example, what happen when optical flow estimation is inaccurate? What happen when content video and style exemplar does not share any similarity in high-level contents?

**Summary Of The Paper:**

This paper presents a novel method for photorealistic video style transfer that conducts color style transfer in videos without undesirable painterly spatial distortions and temporally inconsistent flickering artifacts. The proposed style removal-and-restoration framework, namely ColoristaNet, is capable of learning stylization in a self-supervised fashion. The authors propose to leverage docoupled instance normalization and ConvSLTM units to achieve arbitrary style transfer while keeping the photorealism and temporal conherency. Through a number of comparisons and ablation study, the authors demonstrate ColoristaNet's superior quality to previous state-of-the-art photorealistic style transfer algorithms.

**Summary Of The Review:**

I am leaning towards acceptance. The paper is a technically solid, moderate-to-high impact paper, with no major concerns with respect to evaluation, resources, reproducibility, ethical considerations.

---

### Decision · Program_Chairs · 2023-01-20

**Decision:**

Reject

**Justification For Why Not Higher Score:**

- A complicated pipeline consisting mostly of existing components
- Missing comparisons to relevant related works (specific papers pointed out by two of the reviewers) - in particular, the proposed method seems to do more color transfer than full style transfer and should thus be compared with methods designed for addressing the same task

**Justification For Why Not Lower Score:**

N/A

**Metareview: Summary, Strengths And Weaknesses:**

The paper proposes an approach for style transfer for videos, aiming for improved photorealism and temporal coherence. The method makes use of self-supervised training, a new variant of normalization, as well as LSTM and optical flow for temporal coherence.

Two reviewers suggest rejection, while one leans towards acceptance. The main arguments are as follows.

Pros:
- Good results, fairly convincing experimental evaluation
- The method makes sense and the idea of self-supervised training is interesting

Cons:
- A complicated pipeline consisting mostly of existing components
- Missing comparisons to relevant related works (specific papers pointed out by two of the reviewers) - in particular, the proposed method seems to do more color transfer than full style transfer and should thus be compared with methods designed for addressing the same task

Overall, the paper is not without merit, but given that the paper addresses an established task with many prior works, the experiments would be expected to be very exhaustive and clearly demonstrate that the method indeed usefully contributes on top of existing work. Unfortunately at this point, it is not quite clear, so I recommend rejection.